

# Stability and asymptotic interactions of chiral magnetic skyrmions in a tilted magnetic field

**Bruno Barton-Singer[1]⋆ and Bernd J. Schroers[2]**

**1** Maxwell Institute of Mathematical Sciences and Department of Mathematics,
Heriot-Watt University, Edinburgh EH14 4AS, United Kingdom
**2** Maxwell Institute of Mathematical Sciences and School of Mathematics,
University of Edinburgh, Edinburgh EH9 3FD, United Kingdom

⋆ bsb3@hw.ac.uk

## Abstract

**Using a general framework, interaction potentials between chiral magnetic solitons in a planar system with a tilted external magnetic field are calculated analytically in the limit of large separation. The results are compared to previous numerical results for solitons with topological charge ±1. A key feature of the calculation is the interpretation of Dzyaloshinskii-Moriya interaction (DMI) as a background $SO(3)$ gauge field. In a tilted field, this leads to a $U(1)$-gauged version of the usual equation for spin excitations, leading to a distinctive oscillating interaction profile. We also obtain predictions for skyrmion stability in a tilted field which closely match numerical observations.**

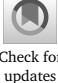

# 1 Introduction

Magnetic skyrmions are examples of topological solitons, topologically non-trivial field configurations of finite energy that minimise an energy functional [1,2]. In general, these configurations are localised in a particular region, and behave like particles in the sense that the lowest-energy excitations of the system involve translating or rotating the localised fields. It is therefore interesting to look at the effective dynamics of these emergent particles, whether that is in the context of second-order Lorentz-invariant dynamics, first-order gradient flow dynamics or first-order Schrödinger dynamics, as it is an important part of the low-energy dynamics of the system.

One may try to understand the dynamics of solitons by reducing the infinite-dimensional dynamics of the field theory to a finite number of degrees of freedom [3], often the positions and orientations of the solitons, and calculating the energy's dependence on these parameters. The assumption is that the dynamics on this finite-dimensional space given by this restricted energy closely approximate the full dynamics on the infinite-dimensional space. Depending on the context, this is known as the moduli space or collective co-ordinate approach. This assumption is also the essence of the Thiele equation [4], where it is assumed that motion of a soliton can be approximated by rigid translation of the statically stable configuration. Generalisations of the Thiele method introduce a finite number of extra parameters [5,6] or allow the shape of the soliton to change depending on position [7].

The accuracy of this approximation has been assessed rigorously for some specific models. Some examples with both first-order Schrödinger dynamics and second-order Lorentz-invariant dynamics are reviewed in [8]. In particular, the moduli space approximation is shown to be justified in certain limits for Chern-Simons vortices evolving according to Schrödinger dynamics [9], a situation which is in some ways analogous to magnetic skyrmion dynamics.

Magnetic skyrmions are topological solitons in a magnetisation field for a specific energy functional modelling chiral magnets [10], and the magnetisation field evolves, in the simplest case, according to the Landau-Lifshitz(-Gilbert) equation [11,12]. Mathematically, this is a combination of gradient flow dynamics, where the field changes so as to decrease the energy functional as fast as possible, and Schrödinger dynamics, where the field evolves in a way that preserves the energy functional. The Schrödinger dynamics comes from the quantum-mechanical evolution of the magnetisation on a microscopic level, which preserves energy, while the gradient flow component represents damping forces that cause energy to leave the system.

An interaction potential is an example of energy restricted to a moduli space in the specific case of two solitons. Conventionally, we subtract the energies of the isolated solitons in order to separate out the potential arising from interaction. In this paper we calculate this interaction potential in the asymptotic limit where the distance between the two solitons becomes large. There is a considerable literature on interaction potentials for solitons. In many cases - for example in the study of nonabelian monopoles [13], nuclear Skyrmions [14–16], baby Skyrmions [17] and abelian vortices [18] - the long-range asymptotics of the interaction

potential can be derived in terms of point sources interacting in a much simpler linear field theory. Such linear approximations can be justified at various levels of rigor in each of those models, but there is no general understanding of when and why linear point-particle pictures capture the asymptotics of soliton interactions.

The question of estimating the interaction energy of magnetic skyrmions analytically was first considered in [19]. More recently, [20,21] calculated the interaction energy in skyrmions supported by frustration and Dzyaloshinskii-Moriya interaction (DMI) respectively. The latter of these drew on methods used, for example, in [17,18]. The interaction potential of magnetic skyrmions in a tilted magnetic field has been numerically simulated [22] and partially calculated [23], but an expression for the analytical potential in terms of the separation of the skyrmions has not been given. The advantages of an explicit formula are several: firstly, it gives new understanding of the interaction potential in terms of multipoles in a $U(1)$ gauge theory which is, in principle, applicable to solitons of any degree. Secondly, it gives an understanding of how the interaction potential depends on the DMI, the external field tilt and the soliton separation, and thirdly it gives us rigorous results, for example whether the interaction energy leads to attraction between solitons for large separations.

The analytical calculation in this paper is based on the following general observations. The field far from the core of any soliton (the 'tail') can be approximated by the solution of the linearised Euler-Lagrange equations, in the presence of a combination of multipole sources (the 'effective source') positioned within the core of the soliton. It follows that to leading order in inverse separation the effective source for the field far away from two well-separated solitons is the sum of the two individual effective sources; this is equivalent to saying that their soliton tails are approximately linearly superposed. The leading contribution to the interaction between two solitons can be expressed in terms of this far field so that, given certain assumptions which are satisfied here, we can approximate the interaction potential in terms of the individual soliton tails (see Appendix B). A surprising feature of the interaction energy of solitons in general is that when it is fully calculated in terms of the effective sources, we often find that it looks like the interaction energy between the effective sources [17,24]: in other words, solitons do not only act like multipoles as sources of their tails, but also respond like those same multipoles to the tail of the other soliton. Here we extend this observation to chiral magnetic solitons in a tilted external field, with the modification that the DMI in general introduces an effective background $U(1)$ gauge field that interacts with the tails, and thus adds a modulation on top of the familiar multipole-multipole interaction energy.

This picture explains the characteristic oscillating behaviour of the interaction energy of magnetic solitons as a function of their separation. What is more, because the interaction formula is derived generally, we can use it to look at other cases of interest. We discuss the interaction of novel textures that have recently been numerically observed in magnetic field applied normal to the plane [21,25,26].

The paper is split into two main sections. In Section 2, we investigate the tails of an individual soliton in a chiral magnet. We first review the interpretation of DMI as a gauge field in Section 2.1, then discuss how this gauge picture changes as magnetisation fields approach the background magnetisation far from the soliton in Section 2.2. We solve the resulting linearised Euler-Lagrange equation in general in Section 2.3, then see how the soliton can be seen as an 'effective source' for its tail in Section 2.4. We then test the accuracy of the linearised Euler-Lagrange equation against numerics in Section 2.5 with details of the numerical methods given in Appendix A. Finally, we note that the behaviour of solutions to the linearised Euler-Lagrange equation gives new insight into elliptical instability of magnetic skyrmions and antiskyrmions in Section 2.6, and compare this to another novel calculation of elliptical instability based on the energy of an isolated $2\pi$-domain wall.

In Section 3.1, we define the interaction potential and describe how it can be approximated in terms of the tails of the individual solitons, under the assumption that these tails fall off exponentially and that the solitons perturb each other a small amount that goes to zero as they go to infinite separation. The details of this calculation are covered in Appendix B. We then substitute in the solutions we have from 2.4 to write the interaction energy in terms of the effective sources of the two solitons in Section 3.2. We consider two cases: in section 3.3, we consider the tilted-field case, where we can compare to numerical observation [22]. This involves the numerical results from Section 2.5. The numerical process is less expensive and more accurate than directly calculating interaction energies, since we only need to simulate an isolated soliton. In Section 3.4, we consider when the applied field is not tilted, simplifying the potential but allowing for a greater diversity of solitons whose interactions have not yet been considered. We derive features of the interaction potential without any numerics. In particular we argue that the presence of 'chiral kinks' [26] on either soliton leads to an attractive force between the solitons.

## 2 Linearised theory of chiral magnets

### 2.1 Energy functional of the chiral magnet

We consider solitons in the magnetisation field $n(\vec{x})$ in the plane, satisfying the constraint $n(\vec{x}) \cdot n(\vec{x}) = 1$. We adopt the convention that boldface vectors $v$ always have three components, and vectors with an arrow $\vec{v}$ always have two. Throughout this paper we are concerned with the chiral magnet energy functional, which we write as

$$E(n) = \int \left( \frac{1}{2} \partial_i n \cdot \partial_i n + D_i \cdot (n \times \partial_i n) + V(n) \right) d^2 x, \qquad (1)$$

where the term $D_i \cdot (n \times \partial_i n)$ is the generalized DMI [27, 28] introduced in different notation in [29] namely $D_i = -\frac{1}{J} \hat{D} e_i$ in terms of the standard orthonormal basis $e_1, e_2, e_3$ of $\mathbb{R}^3$ and the matrix $\hat{D}$, called the DM tensor. Summation over repeated indices with $i = 1, 2$ is assumed throughout. The most commonly considered cases of Bloch-type and Néel-type DMI are obtained by taking $\hat{D}$ to be the identity and a 90 degree rotation about the 3-axis respectively.

We can equally view the chiral magnet energy, or rather a whole family of chiral magnet energies, as an $SU(2)$ or equivalently $SO(3)$-gauged sigma model [30–32]:

$$E(n) = \int \left( \frac{1}{2} |\partial_i n + A_i \times n|^2 + \bar{V}(n) \right) d^2 x. \qquad (2)$$

For later use we note that the gauge transformations are given by a spatially dependent rotation matrix, in axis-angle co-ordinates $R(\theta(\vec{x}), e(\vec{x}))$, under which the fields and DMI transform as follows:

$$\begin{aligned} n &\mapsto \tilde{n} = R(\theta, e)n, \\ A_i &\mapsto \tilde{A}_i = R(\theta, e)A_i - \partial_i \theta e - \sin\theta \, \partial_i e - (1 - \cos\theta) e \times \partial_i e. \end{aligned} \qquad (3)$$

We make some choices in formulating this gauge theory: in [31, 32], the potential term is $-F \cdot n$ where $F$ is the curvature of the gauge field, and thus explicitly gauge-invariant. Here we are more general and let the vector parameters that appear in $\bar{V}(n)$ also rotate under $R(\theta, e)$. We can see that if we expand the energy functional out for a given value of the fields $n$, $A_i$

then we recover a particular chiral magnet model, with

$$\boldsymbol{D}_i = \boldsymbol{A}_i \,,$$
$$V(\boldsymbol{n}) = \bar{V}(\boldsymbol{n}) + \frac{1}{2}|\boldsymbol{A}_i \times \boldsymbol{n}|^2 \,. \tag{4}$$

Therefore different configurations of $\boldsymbol{A}_i$ correspond to different material parameters $\boldsymbol{D}_i$ and $V(\boldsymbol{n})$, and gauge transformations link a configuration $\boldsymbol{n}$ in a system with DMI parameters $\boldsymbol{D}_i = \boldsymbol{A}_i$ to a different configuration $\tilde{\boldsymbol{n}}$ in a different system with DMI parameters $\boldsymbol{D}_i = \tilde{\boldsymbol{A}}_i$ and a different (possibly spatially varying) potential. Although this differs from the usual interpretation of gauge transformations as relating physically equivalent configurations of fields, the language and technology of gauge theory has proved useful in finding exact chiral skyrmion solutions [31], and we will see here that it also provides a conceptually simple way of understanding asymptotic skyrmion interactions.

Configurations of the magnetisation field are classified by their degree, which under suitable assumptions on $\boldsymbol{n}$ is an integer:

$$Q(\boldsymbol{n}) = \frac{1}{4\pi} \int \boldsymbol{n} \cdot (\partial_1 \boldsymbol{n} \times \partial_2 \boldsymbol{n}) d^2 x. \tag{5}$$

We call a non-trivial configuration that minimizes the energy (1) for a given degree a magnetic soliton. Degree $-1$ magnetic solitons are called skyrmions, while degree $+1$ magnetic solitons are called antiskyrmions.

The first term in the energy is the Heisenberg interaction, which favours alignment of $\boldsymbol{n}(\vec{x})$ at a single constant value. Its prefactor can be fixed to $\frac{1}{2}$ by picking appropriate units of energy.

The second term is the DMI. We are most interested in the case where

$$\boldsymbol{D}_1 = -k \begin{pmatrix} \cos\beta \\ \sin\beta \\ 0 \end{pmatrix}, \quad \boldsymbol{D}_2 = -k \begin{pmatrix} -\sin\beta \\ \cos\beta \\ 0 \end{pmatrix}, \tag{6}$$

for real parameters $k$ and $\beta$. We call this axisymmetric DMI, because it is invariant under the physical rotation:

$$\boldsymbol{n}(\vec{x}) \mapsto R(\theta, \boldsymbol{e}_3)\boldsymbol{n}\left(R(-\theta, \boldsymbol{e}_3)\vec{x}\right). \tag{7}$$

Then $\beta = 0$ corresponds to the normal Bloch-type DMI, $k\boldsymbol{n} \cdot (\nabla \times \boldsymbol{n})$, while $\beta = \frac{\pi}{2}$ corresponds to Néel-type DMI. A general angle $\beta$ corresponds to a linear combination of both kinds of DMI.

The third term $V(\boldsymbol{n})$ is the potential function. It attains its minimal value on a set of vacuum configurations. We choose one of them, or take the unique vacuum if the set has only one element, call it $\boldsymbol{n}_0$ in the following and impose it as the boundary value at spatial infinity. Asymptotically we can therefore approximate $\boldsymbol{n}(\vec{x})$ in terms of tangent vector fields to $\boldsymbol{n}_0$. To do this we use the exponential map $\exp_{\boldsymbol{n}(\vec{x})} : T_n S^2 \to S^2$ which takes a tangent vector $\epsilon(\vec{x})$ to the sphere at $\boldsymbol{n}(\vec{x})$ (so $\epsilon(\vec{x}) \cdot \boldsymbol{n}(\vec{x}) = 0$) to the point on the sphere along the great circle in the direction of $\epsilon(\vec{x})$, at a distance $|\epsilon(\vec{x})|$:

$$\exp_{\boldsymbol{n}(\vec{x})}(\epsilon(\vec{x})) = \boldsymbol{n}(\vec{x})\cos|\epsilon(\vec{x})| + \sin|\epsilon(\vec{x})|\frac{\epsilon(\vec{x})}{|\epsilon(\vec{x})|} \,. \tag{8}$$

The map allows us to describe fields $\boldsymbol{n}(\vec{x})$ near $\boldsymbol{n}_0$ in terms of a linear tangent vector field $\boldsymbol{\psi}_n$ to $\boldsymbol{n}_0$, defined via

$$\boldsymbol{\psi}_n(\vec{x}) = \exp_{\boldsymbol{n}_0}^{-1}(\boldsymbol{n}(\vec{x})), \tag{9}$$

and this is essential for our formulation of the linearised theory. The tangent plane $T_{n_0}S^2$ is two-dimensional, so that $\psi_n$ has two coordinates $(\psi_n)_1$, $(\psi_n)_2$ with reference to an orthonormal frame $e_1^n, e_2^n$ oriented so that $e_1^n \times e_2^n = n_0$.

We assume that $V(n)$, expanded around $n_0$, is rotationally symmetric around $n_0$ to quadratic order. Defining $\tilde{V}(\psi_n) = V(\exp_{n_0}(\psi_n))$ this amounts to assuming that

$$\left.\frac{\partial^2 \tilde{V}}{\partial(\psi_n)_i \partial(\psi_n)_j}\right|_{n_0} = \mu^2 \delta_{ij}, \quad \mu^2 \geq 0, \tag{10}$$

i.e. it costs equally to perturb in any direction away from the vacuum. Note that this expression is independent of our choice of basis $e_1^n, e_2^n$.

One physically relevant case where (10) holds is when the potential is the typical combination of a Zeeman interaction from a magnetic field applied normal to the plane and an anisotropy term:

$$V(n) = h_z(1 - n_3) + h_a\left(1 - n_3^2\right), \tag{11}$$

with $h_z + 2h_a \geq 0$. Then $n_0 = e_3$ and $\mu^2 = h_z + 2h_a$. Both $V(n)$ and $n_0$ are symmetric under rotations around $e_3$, so we call such a potential axisymmetric. Another case of interest where (10) holds is when we have just a Zeeman interaction, but from a tilted magnetic field with direction $e_h$:

$$V(n) = h_z(1 - e_h \cdot n). \tag{12}$$

We take $h_z > 0$. Then $n_0 = e_h$ and $\mu^2 = h_z$. Specifically and without loss of generality in the case of axisymmetric DMI (6), we consider a magnetic field tilted at angle $\theta_h$ to $e_3$ so that $e_h = \cos\theta_h e_3 + \sin\theta_h e_1$ and we pick the orthonormal frame $e_1^n = \cos\theta_h e_1 - \sin\theta_h e_3$, $e_2^n = e_2$. In the case of axisymmetric potential (11), we just take $e_1^n = e_1$, $e_2^n = e_2$.

We briefly comment on two cases where the condition (10) is not satisfied. One is where we use the potential (11), but with $h_z + 2h_a < 0$. In that case the minimum of $V(n)$ is attained on a circle and $n_0$ is some particular point on this circle which spontaneously breaks the rotational symmetry of the potential. Thus we call these potentials symmetry-breaking. We now have a zero-mode in one direction away from $n_0$, so that picking a basis $(\psi_n)_1, (\psi_n)_2$ that diagonalises $\left.\frac{\partial^2 \tilde{V}}{\partial((\psi_n)_i \partial(\psi_n)_j)}\right|_{n_0}$ we have diagonal entries

$$\left.\frac{\partial^2 \tilde{V}}{\partial(\psi_n)_1 \partial(\psi_n)_1}\right|_{n_0} = \mu^2, \qquad \left.\frac{\partial^2 \tilde{V}}{\partial(\psi_n)_2 \partial(\psi_n)_2}\right|_{n_0} = 0, \tag{13}$$

where $\mu^2 = -2h_a\left(1 - \left(\frac{h_z}{2h_a}\right)^2\right)$.

The second case is a tilted Zeeman interaction combined with anisotropy, where we have two different non-zero masses:

$$\left.\frac{\partial^2 V}{\partial(\psi_n)_1 \partial(\psi_n)_1}\right|_{n_0} = \mu_1^2, \qquad \left.\frac{\partial^2 V}{\partial(\psi_n)_2 \partial(\psi_n)_2}\right|_{n_0} = \mu_2^2. \tag{14}$$

Both of these cases could still be treated by the same methods we will use below, but the calculations become more complicated because the linearised Euler-Lagrange equations have less symmetry.

## 2.2 Asymptotic form of the chiral magnet energy functional

Soliton solutions can generically be split into two parts: the soliton tail, where the fields are close to the vacuum $n_0$, and the soliton core, which is everywhere else. We expect this to be a small compact region in order to minimize the energy. When we calculate the asymptotic

interaction potential, it is expressed in terms of these soliton tails. So we must approximate these in order to approximate the interaction potential.

We can approximate the tails of an exact solution to the non-linear Euler-Lagrange equations in terms of a corresponding solution to the Euler-Lagrange equations linearised around $n_0$. We will quantify the accuracy of this approximation in terms of the distance from the core below. To find these linearised Euler-Lagrange equations, we expand the energy (1) to quadratic order in $\psi_n$ according to (9):

$$E_{(2)}(\psi_n) = \int \left( \frac{1}{2} \partial_i \psi_n \cdot \partial_i \psi_n + D_i \cdot (n_0 \times \partial_i \psi_n) + D_i \cdot (\psi_n \times \partial_i \psi_n) + \frac{1}{2} \mu^2 |\psi_n|^2 \right) d^2 x, \quad (15)$$

bearing in mind that this energy density and thus the resulting Euler-Lagrange equations are only valid at large $r$, where $\psi_n$ is small. This is equivalent to expanding the Euler-Lagrange equations directly, but helps us see how the DMI becomes a $U(1)$ background gauge field for the tails.

Since $D_i \cdot (n_0 \times \partial_i \psi_n)$ is a divergence, it can be turned into a boundary term which does not affect the Euler-Lagrange equations. It can be removed entirely by a redefinition of the energy as in [30]. For the purpose of determining these equations we therefore ignore this term, but will revisit it when calculating interaction energies.

Further, since $(\psi_n \times \partial_i \psi_n) \parallel n_0$,

$$E_{(2)}(\psi) = \int \left( \frac{1}{2} (\partial \psi)^2 + D_i^{\parallel} \cdot (\psi \times \partial_i \psi) + \frac{1}{2} \mu^2 |\psi|^2 \right) d^2 x + \text{boundary term}, \quad (16)$$

where we split $D_i$ into components parallel and perpendicular to $n_0$, so $D_i = D_i^{\parallel} + D_i^{\perp}$. We also replace $\psi_n$ with $\psi$, as it is a dummy variable at this point.

Defining $a_i = D_i \cdot n_0$, and collecting the components of $\psi$ into complex scalar field, $\psi = \psi_1 + i \psi_2$, this energy can be rewritten as

$$E_{(2)}(\psi) = E_{\text{lin}}(\psi) + \text{boundary term}, \quad (17)$$

where

$$E_{\text{lin}}(\psi) = \int \left( \frac{1}{2} |(\vec{\partial} + i\vec{a})\psi|^2 + \frac{1}{2} (\mu^2 - |\vec{a}|^2)|\psi|^2 \right) d^2 x \quad (18)$$

is the energy functional of a linear field theory for a complex scalar in the background of a fixed abelian gauge field $\vec{a}$. The first term in the energy is a gauged Dirichlet energy for a complex scalar, and the second term is a 'mass term' in the language of quantum field theory. However, unusually and importantly for us, the effective mass $m$ of the scalar is a function of both the potential and the DMI, given by

$$m = \sqrt{\mu^2 - |\vec{a}|^2}. \quad (19)$$

Note that $E_{\text{lin}}$ is only bounded below if $m^2 \geq 0$. If $m^2 < 0$, it can be made arbitrarily negative. This shows that we would be expanding the energy around the wrong vacuum in this case: the state where the field is everywhere equal to the minimum of the potential, $n(\vec{x}) = n_0$, is no longer a minimum of the energy functional due to the effect of the DMI. Thus $m^2 = 0$ corresponds to some sort of phase transition, the details of which will depend on the full nonlinear potential. At $m^2 = 0$, $E_{\text{lin}}$ is bounded below and thus a sensible energy functional viewed in its own right. However we would need to expand the nonlinear energy functional to higher order to see if we are really expanding around the correct vacuum. For the following we will consider parameter regimes where $m^2 > 0$ and thus $m > 0$, but we will return to the significance of the phase transition in Sec. 2.6.

Another way to view the asymptotic abelianisation of the theory is to note that $E_{\text{lin}}$ is the asymptotic form of a nonlinear $U(1)$ gauge theory. To see this, write the energy as

$$E(\boldsymbol{n}) = \int \left( \frac{1}{2}|\partial_i \boldsymbol{n} + A_i^{\parallel} \times \boldsymbol{n}|^2 + \boldsymbol{D}_i^{\perp} \cdot (\boldsymbol{n} \times \partial_i \boldsymbol{n}) + \bar{V}^{\parallel}(\boldsymbol{n}) \right) d^2x \,, \tag{20}$$

where now we can only act with rotations around $\boldsymbol{n}_0$ for our gauge transformations, and the formula for the material parameters of a particular theory reached by our gauge transformations is

$$\boldsymbol{D}_i = \boldsymbol{D}_i^{\perp} + A_i^{\parallel}\,,$$
$$V(\boldsymbol{n}) = \bar{V}^{\parallel}(\boldsymbol{n}) + \frac{1}{2}|A_i^{\parallel}|^2 \,. \tag{21}$$

The potential of this new gauge theory $\bar{V}^{\parallel}(\boldsymbol{n})$ is different is different from both the physical potential $V(\boldsymbol{n})$ and the potential $\bar{V}(\boldsymbol{n})$. Crucially, as long as $m^2 > 0$ it has the same minimum $\boldsymbol{n}_0$ as $V(\boldsymbol{n})$. The term involving $\boldsymbol{D}_i^{\perp}$ does not contribute to $E_{(2)}(\boldsymbol{\psi})$ and thus does not appear in the linearised Euler-Lagrange equations. So the potential $\bar{V}^{\parallel}(\boldsymbol{n})$ is the most meaningful in terms of understanding the soliton tails and the $m^2 = 0$ phase transition.

This phenomenon of asymptotic abelianisation is familiar from the simplest non-abelian Higgs model [33] and underlies the theory of 't Hooft-Polyakov monopoles [34,35], although the context is quite different. Here the gauge field $A_i$ is non-dynamical, so the concept of mass does not apply and the Higgs mechanism has no analogue. We are instead concerned with the fact that the scalar complex field $\psi$, corresponding to the would-be Goldstone bosons of that theory, is acted on by a $U(1)$ gauge field, the massless photon of that theory. In the theory of magnetic skyrmions, most often $\boldsymbol{D}_1^{\parallel} = \boldsymbol{D}_2^{\parallel} = \boldsymbol{0}$, giving a trivial $U(1)$ theory, so this viewpoint has not previously been applied.

Since $\vec{a}$ is constant and a $U(1)$ connection, it is flat and can be 'gauged away' by defining

$$\tilde{\psi} = e^{i\vec{a}\cdot\vec{x}}\psi \,. \tag{22}$$

Then the linearised energy takes the standard form

$$E_{\text{lin}}(\tilde{\psi}) = \int \left( \frac{1}{2}|\vec{\partial}\tilde{\psi}|^2 + \frac{1}{2}m^2|\tilde{\psi}|^2 \right) d^2x \,, \tag{23}$$

where the dependence on $\vec{a}$ is only in the effective mass $m$. This expression for the linearised energy can also be seen as coming from the nonlinear energy functional (20): applying the gauge transformation given by $\boldsymbol{n} \mapsto \tilde{\boldsymbol{n}} = R(\vec{a}\cdot\vec{x}, \boldsymbol{n}_0)\boldsymbol{n}$ eliminates $A_i^{\parallel}$ according to (3), and then expanding the resulting energy to quadratic order around $\tilde{\boldsymbol{n}}_0$ gives us the same final result.

Like the energy expression (18), the equivalent form (23) is only a good approximation to the energy density of a soliton in the asymptotic region far from the centre of the soliton, as represented by large $r$. So solutions of the Euler-Lagrange equation

$$(-\partial_i\partial_i + \mu^2)\psi - 2i\vec{a}\cdot\vec{\partial}\psi = 0\,, \qquad r \text{ large}, \tag{24}$$

are expected to provide good approximations to the soliton tail in the asymptotic region, but not to the soliton core. Using the gauge transformation (22), this becomes the static Klein-Gordon equation

$$(-\partial_i\partial_i + m^2)\tilde{\psi} = 0\,, \qquad r \text{ large}. \tag{25}$$

Thus the gauge transformation described above maps the linearised equation into a standard linear problem whose solutions are well known. In the language of quantum field field theory, it describes static excitations of a scalar field with mass $m$ given by (19).

This is a result of independent interest: we see that spin excitations with mass $\mu$ in the presence of DMI such that $\boldsymbol{D}_i \cdot \boldsymbol{n}_0 \neq 0$ behave like spin excitations with a lower mass $m$ which twist along a direction picked by $\boldsymbol{D}_i \cdot \boldsymbol{n}_0$. This reduction in mass and twisting was observed in [22] for the specific case of Bloch-type DMI and a potential of the form (12), but we see here that this is a general feature of chiral magnets when the DMI and potential have the relation $\boldsymbol{D}_i \cdot \boldsymbol{n}_0 \neq 0$. As we shall see, the simplicity of the linear problem (25) together with the transformation (22) explains remarkably subtle and surprising features of the interaction of magnetic solitons in a tilted field.

Note that asymptotically isotropic potential (10) is crucial for the simplicity of the linearised problem: if we had different 'masses' for perturbing away from the vacuum in different directions, as in (13), (14), the asymptotic potential would not be invariant under rotation around $\boldsymbol{n}_0$. Therefore this gauge twist would still give us an equivalent model where DMI vanishes asymptotically, but the potential would be spatially dependent. The Euler-Lagrange equations can still be asymptotically approximated in this case [36], but the $U(1)$ symmetry which simplifies subsequent calculations has been lost.

## 2.3 Soliton tails in the form of a multipole expansion

We now aim to make concrete the claim at the beginning of Sec. 2.2, that a given nonlinear solution can be approximated by a specific solution of (25). Firstly, given a solution $\boldsymbol{n}$ to the full nonlinear Euler-Lagrange equations, $\boldsymbol{\psi}_n$ is the corresponding linear field satisfying an appropriate nonlinear equation according to (9), leading us to define $\psi_n = (\boldsymbol{\psi}_n)_1 + i(\boldsymbol{\psi}_n)_2$ and $\tilde{\psi}_n = e^{i\vec{a}\cdot\vec{x}}\psi_n$. Our claim is that there is a specific solution to (25), which we call $\tilde{\psi}_n^{\text{lin}}$, that approximates $\tilde{\psi}_n$. In particular $\tilde{\psi}_n^{\text{lin}} \to 0$ at spatial infinity to reflect the fact that $\boldsymbol{n} \to \boldsymbol{n}_0$.

To proceed we look for the general solution to (25), satisfying the same external boundary condition. This is well-known in different coordinate systems. For us, solutions in polar coordinates are particularly interesting because they give rise to an interpretation of a soliton 'from afar' as the source of a linear multipole field. The choice of a centre for our polar coordinate system $(r, \phi)$ is arbitrary at this point, but we will need to discuss this below. Having picked coordinates, we can expand as follows:

$$\tilde{\psi}(r, \phi) = \sum_{M=-\infty}^{\infty} c_M^{\text{lin}}(r) e^{iM\phi}, \tag{26}$$

with $c_M^{\text{lin}}(r)$ a complex function. Substitution into (25) yields the Bessel equation for each $c_M^{\text{lin}}$,

$$\sum_{M=-\infty}^{\infty} \left( c_M^{\text{lin}''}(r) + \frac{1}{r} c_M^{\text{lin}'}(r) - \frac{M^2}{r^2} c_M^{\text{lin}}(r) + m^2 c_M^{\text{lin}}(r) \right) e^{iM\phi} = 0, \qquad r \text{ large}. \tag{27}$$

With the boundary condition that $c_M^{\text{lin}}$ vanishes as $r \to \infty$ and provided $m > 0$, this equation is solved by the modified Bessel function of the second kind of order $M$ [37], that is $c_M^{\text{lin}}(r) = K_M(mr)$ [17]. For $m = 0$, we find $c_0^{\text{lin}}(r) = 0$, $c_{M\neq 0}^{\text{lin}}(r) = r^{-M}$. We need exponential falloff for the derivation below, so we do not consider this case. For large $r$, the leading terms are

$$K_M(mr) = \sqrt{\frac{\pi}{2}} \frac{e^{-mr}}{\sqrt{mr}} + O\left( \frac{e^{-mr}}{(mr)^{\frac{3}{2}}} \right). \tag{28}$$

We can then write the general solution, with $C_M$ arbitrary complex numbers:

$$\tilde{\psi}(\vec{x}) = \sum_{M=-\infty}^{\infty} C_M K_M(mr) e^{iM\phi}. \tag{29}$$

The missing information here that picks out the specific solution $\tilde{\psi}_n^{\text{lin}}$ that we are looking for is the unspecified internal boundary conditions on (25). These reflect the non-linear core of the specific soliton whose tails we want to approximate. To find $\tilde{\psi}_n^{\text{lin}}$, we specify $C_M$ such that if we analogously expand the nonlinear solution

$$\tilde{\psi}_n(r,\phi) = \sum_{M=-\infty}^{\infty} c_M(r)e^{iM\phi}, \tag{30}$$

then $c_M(r) \to C_M K_M(mr)$ as $r \to \infty$, i.e. we set $C_M = \lim_{r\to\infty} \frac{c_M(r)}{K_M(mr)}$, assuming such a limit exists.

At this point we we can quantify the accuracy of approximating $\tilde{\psi}_n$ by $\tilde{\psi}_n^{\text{lin}}$. Above we derived the linearised Euler-Lagrange equations by expanding the energy to quadratic order in $\psi$. Equally, we can take the nonlinear Euler-Lagrange equation

$$\boldsymbol{n} \times \left(-\partial_i\partial_i\boldsymbol{n} + 2\partial_i\boldsymbol{n} \times \boldsymbol{D}_i + \frac{\partial V}{\partial \boldsymbol{n}}\right) = 0, \tag{31}$$

and expand $\boldsymbol{n} = \exp_{\boldsymbol{n}_0}(\boldsymbol{\psi}_n)$ for large radius. If we keep only linear terms in $\boldsymbol{\psi}_n$, then we find Equation (24). This is solved by (29) in general, but only $\tilde{\psi}_n^{\text{lin}}$ as defined above will have the property that $\delta\tilde{\psi} = \tilde{\psi}_n - \tilde{\psi}_n^{\text{lin}}$ is potentially subleading. If we expand to the next order and rewrite in terms of this $\delta\tilde{\psi}$, then we see that

$$(-\partial_i\partial_i + m^2)\delta\tilde{\psi} + O(\tilde{\psi}_n^{\text{lin}})^2 = 0 \implies \tilde{\psi}_n(\vec{x}) = \tilde{\psi}_n^{\text{lin}}(\vec{x}) + O\left(\frac{e^{-2mr}}{mr}\right). \tag{32}$$

The difference between $\tilde{\psi}_n$ and $\tilde{\psi}_n^{\text{lin}}$ is therefore subleading when we substitute it into the interaction energy below.

## 2.4 Soliton tails in terms of an effective source

In the derivation above we solved Equation (25), always bearing in mind that it only applies at large $r$. For what follows it is useful to consider the field $\tilde{\psi}_n^{\text{lin}}$ as defined on the whole plane. This can be done at the cost of introducing a complex 'effective source' $\tilde{\rho}_n$:

$$\left(-\partial_i\partial_i + m^2\right)\tilde{\psi}_n^{\text{lin}} = \tilde{\rho}_n(\vec{x}). \tag{33}$$

This effective source is just a different way of writing the information of an asymptotic solution, and contains no new information. The effective source is introduced because the interaction potential turns out to be written simply in terms of the effective source of one soliton interacting with the tail of the other, which means that the interaction of two solitons can be approximated by the interaction of their effective sources.

We can rewrite the information contained in the constants $C_M$ in terms of the function $\tilde{\rho}_n$. It is useful to define the derivative:

$$D_M = \begin{cases} (\partial_1 + i\partial_2)^M, & M > 0, \\ 1, & M = 0, \\ (\partial_1 - i\partial_2)^M, & M < 0. \end{cases} \tag{34}$$

By using the recurrence relations satisfied by the modified Bessel functions of the second kind [37], we can see that these derivatives move us along the series of independent solutions to (25):

$$D_1\left(K_M(mr)e^{iM\phi}\right) = -mK_{M+1}(mr)e^{i(M+1)\phi},$$
$$D_{-1}\left(K_M(mr)e^{iM\phi}\right) = -mK_{M-1}(mr)e^{i(M-1)\phi}, \tag{35}$$

so in particular $\tilde{\psi}_n^{\mathrm{lin}}(\vec{x})$ is equal to an infinite sum of derivatives acting on $K_0(mr)$. Finally, we note that $\frac{1}{2\pi}K_0(mr)$ is the fundamental solution to this equation, i.e. the solution in the presence of a delta-function source. Using integration by parts, we can therefore write the general solution $\tilde{\psi}_n^{\mathrm{lin}}(r,\phi)$ as the solution to (33) when $\tilde{\rho}_n$ is equal to a combination of sources of the form $D_M\delta(\vec{x})$. These have the interpretation of being idealised multipole sources [38], from which the multipole expansion gets its name.

To be precise, if we write the source as follows:

$$\tilde{\rho}_n(\vec{x}) = 2\pi \sum_{M=-\infty}^{\infty} m^{-|M|} q_M e^{i\gamma_M} D_M \delta(\vec{x}), \qquad (36)$$

with $q_M > 0$, $\gamma_M \in [0, 2\pi)$ real numbers that we can interpret as the strength and orientation of the multipoles respectively, then we can consider the solution to (33) in the presence of this source:

$$
\begin{aligned}
\tilde{\psi}_n^{\mathrm{lin}}(\vec{x}) &= \int \tilde{\rho}_n(\vec{x}')\frac{1}{2\pi}K_0(m|\vec{x}-\vec{x}'|)d^2x' \\
&= \sum_{M=-\infty}^{\infty} (-1)^M m^{-|M|} q_M e^{i\gamma_M} D_M K_0(mr) \\
&= \sum_{M=-\infty}^{\infty} q_M e^{i\gamma_M} e^{iM\phi} K_{|M|}(mr), \qquad (37)
\end{aligned}
$$

and we see that it is equal to (29), with

$$C_M = q_M e^{i\gamma_M}. \qquad (38)$$

So we see that (36) contains all of the data of the general solution to (25). The freedom we had in picking an origin for our polar co-ordinate system manifests itself here as a freedom in choosing where the effective source is located. This can be anywhere in the compact region containing the soliton core. If we choose a different location then the numerical values $q_M$, $\gamma_M$ will change, so multipole moments are always defined relative to a chosen centre.

When solitons have reflection or rotation symmetries it is advantageous to adapt the polar coordinate to these symmetries to simplify the multipole expansion. We have seen one example of symmetry already, for the case of axisymmetric DMI (6). If a soliton is symmetric under the rotation symmetry (7) around a particular point then picking that point as the origin of our co-ordinate system we find that $q_{M\neq1} = 0$.

Axisymmetric DMI has another symmetry, given by reflection of space and the magnetisation at different angles. If we call the reflection about the line at an angle $\phi_0$ relative to the x-axis $P_{\phi_0}$:

$$P_{\phi_0} = \begin{pmatrix} \cos(2\phi_0) & \sin(2\phi_0) \\ \sin(2\phi_0) & -\cos(2\phi_0) \end{pmatrix}, \qquad (39)$$

then this symmetry is given by

$$(\vec{x},(n_1,n_2)) \mapsto \left(P_{\phi_0}\vec{x}, P_{\phi_0+\beta-\frac{\pi}{2}}(n_1,n_2)\right). \qquad (40)$$

Again, a soliton may satisfy this symmetry for a given $\phi_0$ or set of different $\phi_0$. If we pick the origin of our polar co-ordinate system to be on the line of the symmetry, we can impose this symmetry on the multipole expansion (37) to constrain the multipoles, bearing in mind that our answer will depend on our choice of basis vectors of the tangent space $e_1^n$, $e_2^n$ that we made above. For $n_0 = e_3$ and thus $e_1^n = e_1$, $e_2^n = e_2$, we find

$$q_M \neq 0 \implies \gamma_M = -\frac{\pi}{2} + \beta - (M-1)\phi_0 \qquad \mod \pi. \tag{41}$$

In the case of axisymmetric potential (11) and axisymmetric DMI (6), the energy in total has $O(2) \ltimes \mathbb{R}^2$ symmetry. Individual solutions may break it to a discrete subgroup such as $\mathbb{Z}_2 \times \mathbb{Z}_2$ for reflections at two perpendicular lines or $\mathbb{Z}_2$ for a single reflection [26]. In these cases, $\phi_0$ is a free parameter describing the orientation of the soliton and a zero-mode of the energy. In the case of a single $\mathbb{Z}_2$ invariance, (41) tells us how all $\gamma_M$ must change as the orientation of the soliton changes. In the case of two reflection symmetries at right angles, (41) not only sets all $\gamma_M$ but also sets $q_M = 0$ for even $M$. In particular the antiskyrmion in an axisymmetric potential obeys this latter constraint.

Alternatively, solitons may retain the full $O(2)$ symmetry of the energy, as illustrated by skyrmion and skyrmionium solutions [39]. If we centre the polar coordinate system at the fixed point of the spatial rotations, in addition to constraining $q_{M \neq 1} = 0$ it also requires that $\gamma_1 = -\frac{\pi}{2} + \beta$. So solitons like the skyrmion and skyrmionium act as dipole source of fixed orientation for their linear tails.

We can also consider what happens when the energy explicitly breaks axisymmetry, for example in the case of tilted field (12). In this case the energy has only a single $\mathbb{Z}_2$ symmetry, determined by the direction of tilt. For an external magnetic field in the direction $\boldsymbol{e}_h = \sin\theta_h \cos\phi_h \boldsymbol{e}_1 + \sin\theta_h \sin\phi_h \boldsymbol{e}_2 + \cos\theta_h \boldsymbol{e}_3$, one finds $\phi_0 = \phi_h - \beta + \frac{\pi}{2}$. Numerically, skyrmions and antiskyrmions in a tilted field are found to retain this symmetry. As described in Sec. 2.1, if we take $\phi_h = 0$ without loss of generality, we can choose $\boldsymbol{e}_1^n$, $\boldsymbol{e}_2^n$ accordingly, and in this case Equation (41) holds with $\phi_0 = \frac{\pi}{2} - \beta$.

For completeness we note that the real fields $\tilde{\psi}_1$, $\tilde{\psi}_2$ can be approximated by similar multipole expansions in terms of $q_M^1$, $q_M^2$, $\gamma_M^1$, $\gamma_M^2$, subject to the constraint that $q_{-M}^{1,2} = q_M^{1,2}$, $\gamma_{-M}^{1,2} = -\gamma_M^{1,2}$. The terms $q_1^{1,2}$, $q_2^{1,2}$, $q_3^{1,2}$... can be interpreted as the effective dipole, quadrupole, octupole etc. sources for the corresponding real field $\tilde{\psi}_{1,2}$, and $\gamma_M^{1,2}$ as the orientations of these multipoles [38]. The complex sources that we work with are simply linear combinations of these sources $q_M e^{i\gamma_M} = q_M^1 e^{i\gamma_M^1} + iq_M^2 e^{i\gamma_M^2}$, which have no constraint as $\tilde{\psi}$ is a complex field. Equation (41) shows us that the orientations of these multipoles as described by $\gamma_M^{1,2}$ or $\gamma_M$ cannot be directly interpreted as the orientation of the soliton, but they are closely linked.

## 2.5 Numerical calculation of multipole moments

The definition of the multipole moments of a soliton in the previous section relies on a division of the soliton field into a nonlinear core and linear tail, and the results depend on the choice of origin for polar coordinates. The numerical determination of the multipole moments demonstrates both of these features. We illustrate this by considering a skyrmion and an antiskyrmion in the model studied in [22], with a Bloch-type DMI with parameters $\boldsymbol{D}_i = -2\pi\boldsymbol{e}_i$ ($\beta = 0$) and a tilted magnetic field $\boldsymbol{e}_h = \boldsymbol{n}_0 = \sin(\frac{\pi}{3})\boldsymbol{e}_1 + \cos(\frac{\pi}{3})\boldsymbol{e}_3$ of strength $h_z = 0.8 \cdot (2\pi)^2$. This means $\phi_0 = \frac{\pi}{2}$ and $m = 2\pi\sqrt{0.05} \simeq 1.4$, so that the decay length of the soliton tail is $\simeq 0.7$. We choose the origin of our polar coordinate system and thus the location of our multipole sources to be at the point where $\boldsymbol{n}(\vec{x}) = -\boldsymbol{e}_3$. It would be more natural to choose the point at which $\boldsymbol{n} = -\boldsymbol{n}_0$, but throughout this paper we choose $-\boldsymbol{e}_3$ for consistency with the numerical simulations of [22], with which we make quantitative comparison. For either choice, the centre lies on the fixed line of the $\mathbb{Z}_2$ reflection symmetry (40) of the skyrmion and antiskyrmion solution in this model, so we expect to find $\gamma_M$ fixed according to (41). In the following we refer to this point as the 'soliton centre'.

We generate the isolated skyrmion and antiskyrmion by minimising the energy from appropriate initial conditions, see Appendix A for details. To determine the soliton centre, we

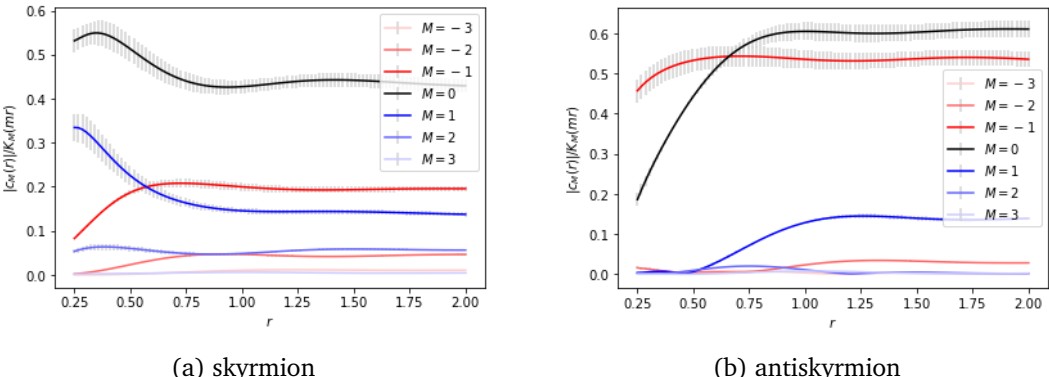

(a) skyrmion                                          (b) antiskyrmion

Figure 1: Magnitude of angular Fourier terms $c_M(r)$ of the tails of the skyrmion and antiskyrmion in a tilted field relative to the Bessel functions $K_M(mr)$, where the inverse decay lengthscale $m$ depends on the DMI strength $k$ and the Zeeman interaction strength $h_z$ through $m = \sqrt{h_z - k^2 \sin^2 \theta_h}$. The deviation for small $r$ is expected, as this represents the nonlinear core of the soliton. The graph should reach a constant value as $r$ becomes larger, giving a measurement of the multipole strengths $q_M$ (36). Error bars come from the uncertainty in the position of the centre of the polar co-ordinate system.

approximate it by the point where $n_3$ is most negative, since $\boldsymbol{n}$ is not exactly equal to $-\boldsymbol{e}_3$ anywhere on the lattice. Using the soliton centre as the origin for polar coordinates $(r, \phi)$ we then invert (30) to get

$$c_M(r) = \frac{1}{2\pi} \int_0^{2\pi} e^{-iM\phi} \tilde{\psi}_{\boldsymbol{n}}(r, \phi) d\phi \, . \tag{42}$$

We know from Equation (32) that $c_M(r) \to c_M^{\text{lin}}(r) = C_M K_M(mr)$ as $r \to \infty$. Thus at sufficiently large $r$, $|c_M(r)|$ should approach a multiple of $K_M(mr)$, and $\arg(c_M(r))$ should be independent of the radius. Moreover, $\gamma_M$ and thus $\arg(c_M(r))$ should be fixed according to (41). However, the uncertainty in the location of the soliton centre, which in reality will not be at the exact point at which $\boldsymbol{n} = -\boldsymbol{e}_3$, will lead to (41) being only approximately satisfied. Theory and numerical results are compared in Figs. 1 and 2, with error bars coming from this uncertainty in the location of the soliton centre, see Appendix A. We only plot comparisons for $-3 \le M \le 3$, since $|c_M(r)|$ is observed to fall off rapidly with increasing $|M|$.

We see that in terms of $|c_M(r)|$, the fit is good beyond $r \simeq 0.8$ for the skyrmion, and $r \simeq 1.25$ for the antiskyrmion. Meanwhile, $\arg(c_M(r))$ is constant all the way down to $r = 0.25$ for some $M$, and even within the core only deviates when the value $|c_M(r)|/K_M(mr)$ is small. There are some $c_M(r)$ for which $|c_M(r)|/K_M(mr) < 0.01$ even for large $r$: these lead to widely varying $\arg(c_M(r))$ and so these are not plotted in Fig. 2. In all cases plotted, the theoretically predicted value of $\gamma_M$ according to (41) is within the errorbars.

## 2.6 Stability of magnetic solitons in a tilted magnetic field

One of the remarkable features of chiral magnetic skyrmions is that, despite their topological nature, they have a variety of instability modes, some of which are studied numerically in [22, 26]. They include elliptic instabilties where skyrmions or antiskrymions elongate indefinitely into domain walls for certain values of the material parameters and the external magnetic field.

When $m^2 < 0$ in (33), it follows that the uniform state $\boldsymbol{n}(\vec{x}) = \boldsymbol{n}_0$ is linearly unstable, as is any solution that approaches $\boldsymbol{n}_0$ in all directions. That is not in itself a good enough reason

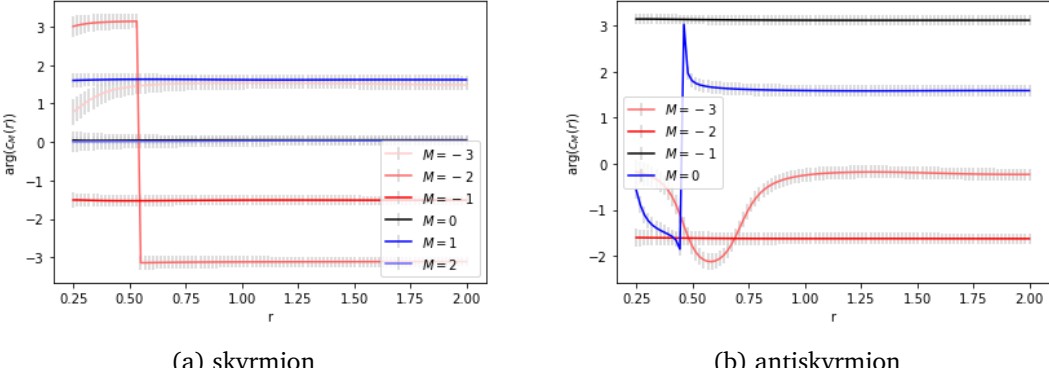

(a) skyrmion  (b) antiskyrmion

Figure 2: Argument of the angular Fourier terms $c_M(r)$ of the tails of the skyrmion and antiskyrmion in a tilted field. Error bars come from the uncertainty in the position of the centre of the polar co-ordinate system. We exclude those $c_M(r)$ where the absolute value is so small that the error in the argument becomes $O(2\pi)$. The lines plotted should approach a constant at large $r$ where the linear approximation is justified, while at small $r$ deviation is expected. If the soliton respects the reflection symmetry (40), then (41) further tells us the value of this constant up to $\pm\pi$. This graph thus acts as both a test of the linearised Euler-Lagrange equations approximation and a check that numerically found skyrmions and antiskyrmions in a tilted field have the reflection symmetry described.

to think that this region tells us anything about elliptical instability. However, as we approach this region, the lengthscale of decay diverges and so the whole nonlinear solution, while it exists, will expand. At the same time, the oscillation along a given direction that is favoured by the DMI as described by $\vec{a}$ will therefore become more pronounced. These are suggestions that before or at the same time as this phase transition, solitons become elliptically unstable.

We therefore plot the $m^2 < 0$ region and compare it to numerical results [22] for the onset of elliptical instability. In the case of axisymmetric DMI with strength $k$ (6) and a tilted applied field with strength $h_z$ and tilt $\theta_h$ as in (12), $m^2 < 0$ is equivalent to $h_z < k^2 \sin^2 \theta_h$. In Fig. 3 it is is plotted as a grey region bounded by a black solid line, along with the numerical data for the onset of elliptical instability for skyrmions and antiskyrmions in red and blue crosses respectively.

In the numerics $k = 1$ and the DMI is Bloch-type, i.e. $\beta = 0$, but this leads to no loss of generality as any axisymmetric DMI is equivalent to Bloch under rotation of $\boldsymbol{n}$ and by defining suitable units of length $k$ can be set to 1. Similarly, in the numerics only $Q = -1$ solutions are considered. However, there is a transformation

$$
\begin{aligned}
n_3 &\mapsto -n_3 \,, \\
\vec{x} &\mapsto -\vec{x} \,, \\
\theta_h &\mapsto \pi - \theta_h \,,
\end{aligned}
\tag{43}
$$

which leaves the energy of a solution invariant while changing the sign of its charge, $Q \mapsto -Q$. Since this transformation changes $\theta_h$ it is not a symmetry of the energy but a transformation that links equal-energy solutions in different systems, like the gauge transformation introduced in Sec. 2.1. This transformation interchanges the role of skyrmion and antiskyrmion as $\theta_h$ is varied from 0 to $\pi$, as noted in [22]. It allows us to use the same numerical data to insert the points at which $Q = 1$ solutions experience elliptical instability.

The lower bound for instability turns out to closely fit the numerical data for applied fields tilted closer to the plane, while for fields close to the perpendicular, the lower bound is not very

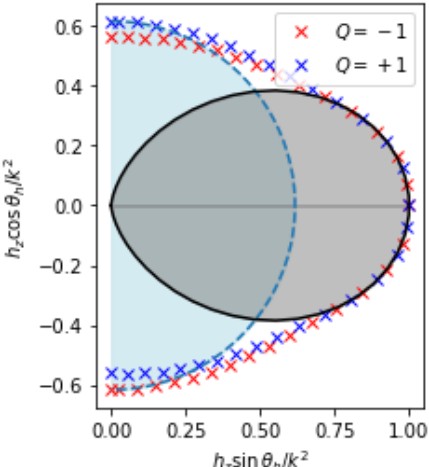

Figure 3: Comparison of theoretical predictions and numerical observations of the onset of soliton instability in a tilted magnetic field, with strength $h_z$, tilt $\theta_h$, DMI strength $k$. The blue shaded area inside the dashed blue line is the region of the $(h_z, \theta_h)$ phase diagram where domain walls have negative energy per unit length. The grey shaded area inside the solid black line is the region where the vacuum is linearly unstable. These give two theoretical estimates for the onset of instability. Blue and red crosses show the numerically observed values of $(h_z, \theta_h)$ at which $Q = \pm 1$ solutions (related by the transformation (43)) become elliptically unstable as $h_z$ is decreased [22].

useful; there is a large region of the phase diagram for which elliptical instability happens but is not seen by the calculation above. To understand the instability in this region, we use an energy comparison with domain walls, generalising the method employed [26, 40] for axisymmetric potential (11) to include tilted fields.

If, without loss of generality, we take Bloch-type DMI, that is axisymmetric DMI (6) with $\beta = 0$, and tilted potential (12), we can construct the domain wall ansatz

$$\boldsymbol{n}(x) = \begin{pmatrix} \sin(\theta_h + f(x_2)) \\ 0 \\ \cos(\theta_h + f(x_2)) \end{pmatrix}, \tag{44}$$

where $f(x_2)$ goes from 0 to $2\pi$ as we go from $-\infty$ to $+\infty$. This orientation is chosen to give a negative contribution to the DMI. This domain wall, if we do not specify this orientation, is the most general solution approaching $\boldsymbol{n}_0$ at $x_2 \to \pm\infty$ retaining the symmetries of the energy $x_1 \mapsto x_1 + a$ and $(x_1, n_2) \mapsto (-x_1, -n_2)$ (40), and thus by the principle of symmetric criticality [41] if we minimise the energy of this ansatz with respect to $f$ we will find a true stationary point of the energy. For tilted field and axisymmetric DMI we can always use the combination of translation and a reflection-like symmetry to have an ansatz depending on a single function $f$, but as a result of the loss of the $O(2)$ symmetry such an ansatz constrains us to consider domain walls lying parallel to some axis. For this choice of DMI and tilt it is the $x_1$ axis, so we are specifically investigating if solitons will extend along the $x_1$ direction by assuming their cross-section in the $x_2$ direction is well-approximated by an isolated domain wall.

Substituting (44) into the energy, we get the Sine-Gordon energy modified by a boundary term coming from the DMI, just as in the case of normally applied magnetic field without

anisotropy:

$$E_{\min}(f) = \left( \int_{-\infty}^{\infty} dx_1 \right) \left( -2\pi k + 8\sqrt{h_z} \right). \tag{45}$$

Hence the energy per unit length of such a domain wall changes sign at $h_z = \left( \frac{k\pi}{4} \right)^2$. For smaller $h_z$, the domain wall has negative energy per unit length. This suggests an elliptical instability, because any soliton necessarily has a cross-section that looks like the domain wall above, and if domain walls have negative energy per unit length then we expect them to grow in length.

The instability line $h_z = \left( \frac{k\pi}{4} \right)^2$ is also plotted in Fig. 3 as a dashed blue line, with the shaded blue region corresponding to $h_z < \left( \frac{k\pi}{4} \right)^2$. We see that the domain wall instability calculation, by contrast with the $m^2 = 0$ curve, is most useful as $\theta_h$ approaches 0 or $\pi$. In the previous calculations for axisymmetric potential [26], it was shown that the curve $E_{\min}(f) = 0$ is a particularly good fit to numerical observations of elliptical instability for antiskyrmions, where the cross-section perpendicular to the 'long' axis of the antiskyrmions is well approximated by an isolated domain wall. Following the $Q = -1$ solution as we vary $\theta_h$ from 0 to $\pi$ produces a soliton that is effectively an antiskyrmion, in the sense that it is in correspondence with the antiskyrmion solution at $\theta_h = 0$ under the transformation (43). This antiskyrmion-like solution is oriented so that its cross-section along $x_2$ is a good approximation to an isolated domain wall. Therefore we see this good fit as $\theta_h \to \pi$. Meanwhile, in normal magnetic field the domain wall method slightly overestimates critical $h_z$, as some energy barrier separates the axisymmetric skyrmion from an arbitrarily extended $Q = -1$ solution. We see this overestimation emerge as $\theta_h \to 0$. The reverse applies for the $Q = +1$ solutions.

## 3 Interaction potential for magnetic solitons

### 3.1 The interaction potential in terms of soliton tails

As described in the introduction, to define an interaction potential of solitons we must first describe how we construct a suitable moduli space of two interacting solitons. Given a field $\boldsymbol{n}^A$ describing soliton $A$ and a field $\boldsymbol{n}^B$ describing soliton $B$, we must construct a field $\boldsymbol{n}^{AB}$ that describes soliton $A$ and $B$ located at points $\vec{R}^A$, $\vec{R}^B$ in the plane. There is no canonical way to do this, or even say what it means for a soliton to be 'located' at a particular point. In the following calculation we do not in fact commit to a particular method, but for concreteness it is useful to have a particular procedure in mind, so that we can see if our assumptions are reasonable.

Here we take inspiration from the way interaction potentials are defined numerically [7, 22, 42], which we here call pinning: in this procedure the interaction energy of two skyrmions located at $\vec{R}^A$, $\vec{R}^B$ is taken to be the minimum energy configuration reached by gradient descent from some appropriate starting ansatz, subject to the constraint that the lattice points at $\vec{R}^A$, $\vec{R}^B$ are fixed to a particular value within the soliton core, generally $-\boldsymbol{e}_3$. We call the point within a skyrmion where this value is attained the soliton centre in what follows. We can think of this as an adiabatic approximation, where under gradient descent the timescale of solitons moving together or apart is much larger than all other timescales, so the solitons instantaneously adjust to minimise their energy at a given separation.

We suppose that we can analytically define a moduli space in an analogous way, with some modifications. Instead of gradient descent, which depends on our choice of starting ansatz, we define $\boldsymbol{n}^{AB}$ as the absolute minimiser over all configurations satisfying pinning constraints which approximate solitons $A$ and $B$ having their centres at points $\vec{R}^A$, $\vec{R}^B$. For skyrmions in axisymmetric potential, the point of rotational symmetry makes a natural choice for soliton

centre. For general solitons, the choice is more arbitrary. Given some assumptions we will discuss below, the interaction potential we calculate is independent of the choice of centres, although different choices of centre will lead to the same potential being described in terms of a different independent co-ordinate. Since we compare to numerics where $\boldsymbol{n} = -\boldsymbol{e}_3$ is chosen as the centre, we must pick that here as well. Again in contrast to skyrmions in axisymmetric potential, general solitons may have internal degrees of freedom such as orientation, represented here by (possibly multi-component) quantities $\phi^A, \phi^B$. In such cases, we must refine the procedure by adding constraints to fix these parameters also. In theories with a translationally invariant energy expression we can choose $\vec{R}^A = \vec{0}$ and write simply $\vec{R}$ for $\vec{R}^B$, and we do this here.

With all this in mind, we define the configuration $\boldsymbol{n}^{AB}[\vec{R}; \phi^A, \phi^B]$, which models the interactions of solitons $A$ and $B$ described by fields $\boldsymbol{n}^A$, $\boldsymbol{n}^B$, determined by internal parameters $\phi^A$, $\phi^B$, and with charges $Q^A, Q^B$, as the absolute minimiser within the space of $Q^A + Q^B$ configurations where $\boldsymbol{n}(\vec{0}) = \boldsymbol{n}(\vec{R}) = -\boldsymbol{e}_3$, and any further necessary constraints to determine $\phi^A$, $\phi^B$ and rule out other pairs of interacting solitons with the same total charge. Our moduli space modelling the interaction of these two solitons is then obtained by allowing $\vec{R}$ to vary over a suitable open set typically of the form $\{\vec{R} \mid |\vec{R}| > R_c\}$, and the internal degrees of freedom to vary arbitrarily. The interaction potential on this moduli space is defined as

$$V^{AB}\left(\vec{R}; \phi^A, \phi^B\right) = E\left(\boldsymbol{n}^{AB}\right) - E\left(\boldsymbol{n}^A\right) - E\left(\boldsymbol{n}^B\right). \tag{46}$$

We now derive an asymptotic expression for the interaction potential in the limit of large soliton separation, that is $R = |\vec{R}| \gg \frac{1}{m}$. Motivated by the analogy to the numerical procedure, we assume that as $R$ becomes large, $\boldsymbol{n}^{AB}$ approaches the field $\boldsymbol{n}^A$ near $\vec{0}$, and the field $\boldsymbol{n}^B$ near $\vec{R}$. We also assume that as we go far from both $\vec{0}$ and $\vec{R}$, the field $\boldsymbol{n}^{AB}$ approaches linear superposition of the tails of the two solitons, as discussed in the introduction. Finally, we restrict ourselves to consider local energy functionals with the property that $\boldsymbol{n}^A$, $\boldsymbol{n}^B$ and their derivatives fall off exponentially towards the vacuum away from their respective centres. Note that this last assumption is satisfied in the case considered in Sec. 2.3, with the decay lengthscale given by $\frac{1}{m}$ (29). These are the central assumptions to the results that follow.

The formula we obtain is remarkably general and simple, and is the main result of this paper. However, the derivation is rather technical and therefore relegated to Appendix B. The calculation involves dividing the plane into two infinite parts, $\sigma^A$ and $\sigma^B$, which respectively contain the core of soliton $A$ and the core of soliton $B$. The dividing curve is thus called $\partial\sigma^A$, see Fig. 4.

The result of the calculation is an approximation for the interaction potential purely in terms of the soliton tails $\boldsymbol{\psi}_{\boldsymbol{n}^A}, \boldsymbol{\psi}_{\boldsymbol{n}^B}$:

$$V^{AB}\left(\vec{R}; \phi^A, \phi^B\right) = 2\int_{\partial\sigma^A} \boldsymbol{\psi}_{\boldsymbol{n}^B} \cdot \left(\partial_i \boldsymbol{\psi}_{\boldsymbol{n}^A} + \boldsymbol{D}_i \times \boldsymbol{\psi}_{\boldsymbol{n}^A}\right) dS^i + O\left(e^{-\frac{3}{2}mR}\right), \tag{47}$$

where $dS^i$ represents the vector surface element of $\partial\sigma^A$. Although $\partial\sigma^A$ appears in this expression, its exact form and location do not matter, as we shall see below.

We can rewrite this in terms of the complex field $\psi_{\boldsymbol{n}} = (\psi_{\boldsymbol{n}})_1 + i(\psi_{\boldsymbol{n}})_2$ and $a_i = \boldsymbol{D}_i \cdot \boldsymbol{n}_0$ as defined in (18) to get

$$V^{AB}(\vec{R}; \phi^A, \phi^B) = 2\Re \int_{\partial\sigma^A} \left(\bar{\psi}_{\boldsymbol{n}^B}(\vec{\partial} + i\vec{a})\psi_{\boldsymbol{n}^A}\right) \cdot d\vec{S} + O\left(e^{-\frac{3}{2}mR}\right). \tag{48}$$

Moreover, from (32), $\psi_{\boldsymbol{n}^A}, \psi_{\boldsymbol{n}^B}$ can be replaced by $\psi_{\boldsymbol{n}^A}^{\text{lin}}, \psi_{\boldsymbol{n}^B}^{\text{lin}}$ to the level of approximation we are already working at, meaning we can describe the leading behaviour of the potential in terms of the linearised soliton tails:

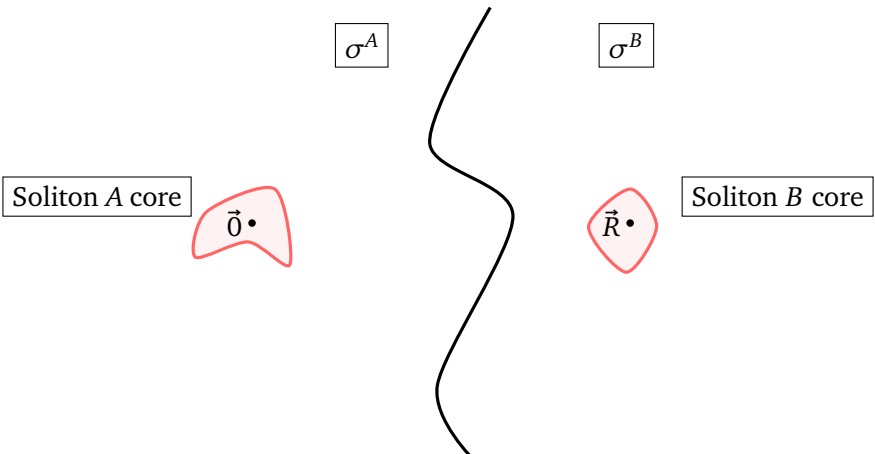

Figure 4: Schematic representation of the setup for calculating the interaction potential.

$$V^{AB}(\vec{R}; \phi^A, \phi^B) = V_{\text{lin}}^{AB}(\vec{R}; \phi^A, \phi^B) + O\left(e^{-\frac{3}{2}mR}\right), \tag{49}$$

where

$$V_{\text{lin}}^{AB}(\vec{R}; \phi^A, \phi^B) = 2\Re \int_{\partial\sigma^A} \left(\bar{\psi}_{n^B}^{\text{lin}}(\vec{\partial} + i\vec{a})\psi_{n^A}^{\text{lin}}\right) \cdot d\vec{S}. \tag{50}$$

Equation (48) is the $U(1)$-gauged version of the interaction energy of complex scalar fields in a linear theory. This is analogous to the interaction potential of baby skyrmions [17], where the linear interaction potential is equivalent to the interaction energy of complex scalar fields in a linear theory without a gauge field, and thus equivalent to the independent interaction of two real scalar fields.

Equation (50) tells us that if $\psi_{n^A}^{\text{lin}}$ or $\psi_{n^B}^{\text{lin}}$ winds around the origin in the complex plane as we vary either $\vec{R}$ or some of the $\phi^A$, $\phi^B$, then $V_{\text{lin}}^{AB}$ must change sign. We will use this fact below to determine when attraction can exist between magnetic solitons.

We also see that we have a natural expression given the interpretation of $\vec{a}$ as a $U(1)$ gauge field:

$$V_{\text{lin}}^{AB}(\vec{R}; \phi^A, \phi^B) = 2\Re\left(e^{-i\vec{a}\cdot\vec{R}} \int_{\partial\sigma^A} \left(\bar{\tilde{\psi}}_{n^B}^{\text{lin}} \vec{\partial} \tilde{\psi}_{n^A}^{\text{lin}}\right) \cdot d\vec{S}\right). \tag{51}$$

Again, we see that this is just like the interaction energy without DMI, but with a gauge twist applied to the fields, and with a modulation representing the parallel transport between the two soliton centres with respect to the gauge field $\vec{a}$. Integrating by parts in either the bulk $\sigma^A$ or $\sigma^B$ and using the fact that $\tilde{\psi}_{n^A}^{\text{lin}}$, $\tilde{\psi}_{n^B}^{\text{lin}}$ solve (33) with corresponding effective sources $\tilde{\rho}_{n^A}$, $\tilde{\rho}_{n^B}$, we can write (51) also as bulk integrals:

$$V_{\text{lin}}^{AB}(\vec{R}; \phi^A, \phi^B) = -\Re\left(e^{-i\vec{a}\cdot\vec{R}} \int_{\sigma^A} \bar{\tilde{\psi}}_{n^B}^{\text{lin}} \tilde{\rho}_{n^A} d^2x\right) = \Re\left(e^{-i\vec{a}\cdot\vec{R}} \int_{\sigma^B} \tilde{\psi}_{n^A}^{\text{lin}} \bar{\tilde{\rho}}_{n^B} d^2x\right). \tag{52}$$

This is like the interaction energy in [17], but before we take the real part of this complex multipole interaction, we first multiply it by the parallel transport factor $e^{-i\vec{a}\cdot\vec{R}}$, complicating a representation in terms of real multipoles. Moreover, without the high symmetry of that scenario, we cannot reduce it to solely an interaction of dipoles. However, we do see the phenomenon discussed in the introduction: not only do solitons act as a combination of multipole

sources for their tails, which can be thought of as just as a convenient mathematical representation, but they also respond like those same multipole sources to the presence of another soliton. Note the irrelevance of the exact location of the boundary $\partial \sigma^A$, given that the sources $\tilde{\rho}_{\boldsymbol{n}^A}$, $\tilde{\rho}_{\boldsymbol{n}^B}$ are located at $\vec{0}$, $\vec{R}$.

The advantage of this representation is that, by substituting in Equations (36), (37) it will allow the interaction to be explicitly written in terms of the separation of the two multipole sources. However, as discussed in Sec. 2.4, these multipoles are only meaningful given a particular choice of centre of expansion. The final expression will be written in terms of the separation between the centres that we choose, with multipoles that will be different for different choices of centre. Therefore for different choices of centre, $V_{\text{lin}}^{AB}$, expressed as a function of the separation between the multipoles (which is itself a function of $\vec{R}$) could look quite different. However, the expression (51) for $V_{\text{lin}}^{AB}(\vec{R})$ is fundamentally in terms of $\tilde{\psi}_{\text{lin}}$ and thus independent of the choice we make.

The expression (51) is valid without constraint on $\boldsymbol{D}_i$ and $V(\boldsymbol{n})$, provided the tails fall off appropriately, so could be used more generally. However, as discussed above, the solutions to the linearised Euler-Lagrange equations for general $V(\boldsymbol{n})$ are more complicated.

In [23], the authors substitute numerical solutions for the tails of specific solitons at given couplings into (48), for a variety of energy functionals. However, here we continue to an explicit expression for $V_{\text{lin}}^{AB}$, which has certain advantages as discussed in the introduction. We also use numerical simulation of an isolated soliton at specific couplings when we compare to numerical results for the interaction potential, but even in this case the approach is more general: a single numerical simulation of each isolated soliton will give us enough information to approximate the interaction potential between any two solitons at arbitrary $\vec{R}$.

## 3.2 The interaction potential in terms of effective sources

We can now explicitly calculate the interaction potential. We consider two solitons, $A$ and $B$, described by corresponding sources $\tilde{\rho}_{\boldsymbol{n}^A}$ and $\tilde{\rho}_{\boldsymbol{n}^B}$ located at $\vec{0}$ and $\vec{R}$ respectively:

$$\tilde{\psi}_{\boldsymbol{n}^A}^{\text{lin}}(\vec{x}; \phi^A) = \sum_M q_M^A e^{i(M\phi + \gamma_M^A)} K_{|M|}(mr),$$

$$\tilde{\psi}_{\boldsymbol{n}^B}^{\text{lin}}(\vec{x}; \phi^B) = \sum_M q_M^B e^{i(M\phi(\vec{x}-\vec{R}) + \gamma_M^B)} K_{|M|}\left(m|\vec{x}-\vec{R}|\right),$$

(53)

where $\phi(\vec{x}-\vec{R})$ is the polar angle in co-ordinates centred on $\vec{R}$, and $q_M^{A,B}$, $\gamma_M^{A,B}$ are functions of $\phi^{A,B}$. This dependence can be constrained from symmetry, which we will see below.

In the cases where $q_M^{A,B}$, $\gamma_M^{A,B}$ are not constrained enough from symmetry, we find them numerically from the asymptotics of a single soliton, using the results from Sec. 2.5. Note that this is why we chose to make our multipole expansion around the point at which $\boldsymbol{n} = -\boldsymbol{e}_3$ in that section: the interaction potential is ultimately written in terms of the separation between the multipole sources, while it is defined as a function of $\vec{R}$, the separation between the soliton centres (as well as internal degrees of freedom). To be able to write our potential explicitly, these two separations must be the same, so the multipole sources must be chosen to lie at the soliton centres.

We now substitute this expansion into the integral in (52), using the relation between the expansion of a field and the expansion of its source expressed in equations (36), (37):

$$\int_{\sigma^A} \tilde{\tilde{\psi}}_{\boldsymbol{n}^B}^{\text{lin}} \tilde{\rho}_{\boldsymbol{n}^A} d^2 x = 2\pi(-1)^N \sum_{M,N} q_M^A q_N^B e^{i(\gamma_M^A - \gamma_N^B)} K_{|M-N|}(mR) e^{i(M-N)\chi},$$

(54)

and thus

$$V_{\text{lin}}^{AB}(\vec{R}; \phi^A, \phi^B) = 2\pi \sum_{M,N} (-1)^{N+1} q_M^A q_N^B \cos\left(\gamma_M^A - \gamma_N^B - \vec{a}\cdot\vec{R} + (M-N)\chi\right) K_{|M-N|}(mR), \quad (55)$$

where from $\vec{a}$ defined in (18) we define $a = |\vec{a}|$ and $\chi$ as the angle between $\vec{R}$ and $\vec{a}$.

This is the interaction potential between solitons in a chiral magnet in large generality: as described above, we have assumed that the potential is asymptotically isotropic about the vacuum (10), general DMI, $m$ as defined in (19) is real, and no other interactions. Because of the independence of which soliton we take at $\vec{0}$, this potential has the symmetry $\chi \to \chi + \pi$, $A \leftrightarrow B$. This means for interactions between like solitons, $\chi \to \chi + \pi$ is a symmetry.

At sufficiently large $R$, we can use the expansion of $K_M(mR)$ in powers of $\frac{1}{mR}$ given in Equation (28):

$$V_{\text{lin}}^{AB}(\vec{R}, \phi_0^A, \phi_B^0) = \sqrt{2\pi^3} f(\chi; \phi_0^A, \phi_0^B) \frac{e^{-mR}}{\sqrt{mR}} + O\left(\frac{e^{-mR}}{(mR)^{\frac{3}{2}}}\right), \quad (56)$$

where

$$f(\chi; \phi_0^A, \phi_0^B) = \sum_{M,N} (-1)^{N+1} q_M^A q_N^B \cos(\gamma_M^A(\phi_0^A) - \gamma_N^B(\phi_0^B) - aR\cos\chi + (M-N)\chi). \quad (57)$$

At this radius, we can similarly expand the individual soliton tails:

$$\tilde{\psi}_{n^{A,B}}^{\text{lin}}(\vec{x}; \phi^{A,B}) = \left(\sum_M q_M^{A,B} e^{i(M\phi + \gamma_M^{A,B})}\right) \sqrt{\frac{\pi}{2}} \frac{e^{-mr}}{\sqrt{mr}} + O\left(\frac{e^{-mr}}{(mr)^{\frac{3}{2}}}\right), \quad (58)$$

from which it follows that $V_{\text{lin}}^{AB}$ can be approximated in terms of a product of the two soliton tails at the midpoint between them:

$$V_{\text{lin}}^{AB}(\vec{R}; \phi^A, \phi^B) = -\psi_{n^A}\left(\frac{R}{2}, \chi; \phi^A\right) \cdot \psi_{n^B}\left(\frac{R}{2}, \chi + \pi; \phi^B\right) \sqrt{2\pi mR} + O\left(\frac{e^{-mR}}{(mR)^{\frac{3}{2}}}\right). \quad (59)$$

This is a remarkably simple form of the interaction energy, if one wants to quickly estimate whether two solitons will attract or repel at a large distance, but for the rest of the paper we will continue to use the expression (55) for greater accuracy at smaller $R$.

## 3.3 Comparison to numerically calculated interaction potential in a tilted field

In the case of tilted field (12), there are two known stable solitons, the skyrmion and anti-skyrmion [22]. These are known to both retain the single $\mathbb{Z}_2$ symmetry of the energy that remains for tilted field, of the form (40) with $\phi_0$ fixed by the parameters of the theory, and have no internal degrees of freedom. Therefore to construct our moduli space, we do not need to add any constraints besides fixing $n^{AB}(\vec{0}) = n^{AB}(\vec{R}) = -e_3$ and specifying the overall topological charge as $-2$, $0$ or $2$. According to (41), $\gamma_M^A$ and $\gamma_M^B$ are fixed to the same value such that $\gamma_M^A - \gamma_N^B = (N-M)\phi_0$. Without loss of generality, we consider Bloch-type DMI (6), so $\beta = 0$, and field tilt in the direction of $e_1$, as discussed in Sec. 2.1, so that $\phi_h = 0$, $\beta = 0$ and thus $\phi_0 = \frac{\pi}{2}$, so we have

$$V_{\text{lin}}^{AB}(R, \chi) = 2\pi \sum_{M,N} (-1)^{N+1} q_M^A q_N^B \cos\left((N-M)\frac{\pi}{2} - aR\cos\chi + (M-N)\chi\right) K_{|M-N|}(mR). \quad (60)$$

Once $\gamma_M^A$, $\gamma_N^B$ satisfy this, it imposes a corresponding symmetry on the potential of $\chi \to \pi - \chi$. Meanwhile, we calculate $q_M$ by using the results from Sec. 2.5, which only

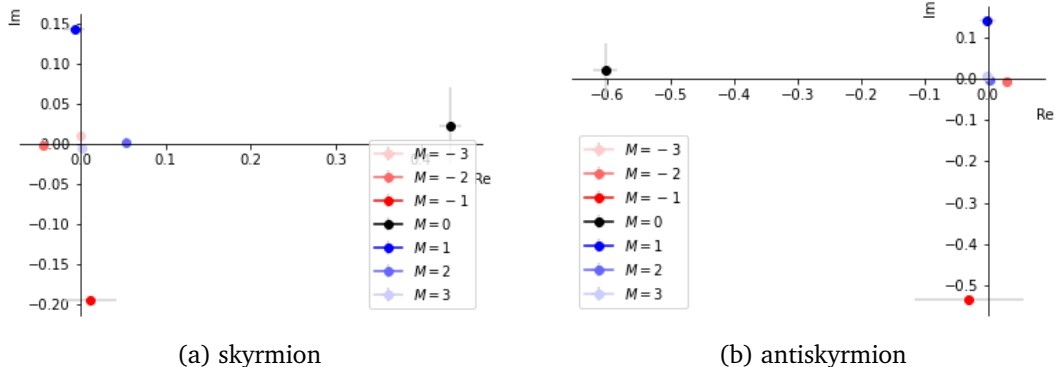

|                | (a) skyrmion | (b) antiskyrmion |

Figure 5: Plotting the numerically extracted multipole sources $C_M$ seen in the general solution of the linearised Euler-Lagrange equations for solitons in a tilted field, $\tilde{\psi}_{\text{lin}} = \sum_{M=-\infty}^{\infty} C_M e^{iM\phi} K_M(mr)$, where the inverse decay lengthscale $m = \sqrt{h_z - k^2 \sin^2 \theta_h}$ for DMI strength $k$, Zeeman interaction strength $h_z$.

apply to the specific material parameters considered: $h_z = 0.8 \cdot (2\pi)^2$, $k = 2\pi$. While this means that even in our analytical approach some numerics is required, the cost of simulating a single soliton in isolation is much lower than calculating the interaction directly, which requires pinning the soliton at every possible distance from every other soliton it could potentially interact with. Moreover, our interaction energy can be calculated for arbitrarily large $R$, while numerical error becomes dominant in the direct interaction calculation as the energy difference from isolated solitons becomes exponentially small.

To calculate $q_M = |C_M|$, we fit our numerically found $|c_M(r)|$ to $|C_M| K_M(mr)$. To find $\gamma_M = \arg(C_M)$, we take the angular mean of $\arg(c_M(r))$. Both the fitting and the mean are taken between $r = 1$ and $r = 2$, so as to exclude the nonlinear soliton core. The errors on $\arg(c_M(r))$ and $|c_M(r)|$ are propagated through to errors in $|C_M|$, $\arg(C_M)$, bearing in mind that the errors are not independent at each $r$. The resulting $C_M$ and errorbars are plotted in Fig. 5.

When $\theta_h = \frac{\pi}{2}$ there is a symmetry of the energy $n_3 \mapsto -n_3$, $\vec{x} \mapsto -\vec{x}$ interchanging skyrmion and antiskyrmion solutions [22], which can be seen as a special case of the transformation discussed in Sec. 2.6, and as a result skyrmion and antiskyrmion sources are related by a reflection around the imaginary axis. In Fig. 5 we see that the symmetry still holds approximately, suggesting that the values of the sources change continuously with respect to the material parameters.

Having numerically found $q_M$, $\gamma_M$, for both skyrmion and antiskyrmion, we can substitute them into (55) and plot $V_{\text{lin}}(R, \chi)$, comparing it to the numerical results in [22]. Because that paper defined $\vec{R}$ as the distance between the two points where $\boldsymbol{n}$ is fixed to equal $-\boldsymbol{e}_3$, we have defined our multipole locations to be in the same place, as discussed above.

Our result is only asymptotically valid, while the data is at small distances, but there is nevertheless good agreement for $R > 1.5$, see Fig. 6. We see that because of the error in $\gamma_M$ discussed in Sec. 2.5, there is some small violation of the $\chi \to \pi - \chi$ symmetry in the analytical prediction, which could be removed by using the theoretically predicted values of $\gamma_M$.

This analysis thus explains the observed oscillation of the inter-soliton potential in a tilted field as a manifestation of the emergent $U(1)$ gauge theory (18). In general it is fixed by the DMI lengthscale but with a non-trivial factor $\frac{1}{\sin \theta_h}$. As we take the tilt angle to 0, we would see the period of the oscillation diverge to infinity.

We also have an explanation for the unusual asymmetric lobe structure of the skyrmion-antiskyrmion interaction potential, by analogy to the Roget's palisade illusion [43], or rolling

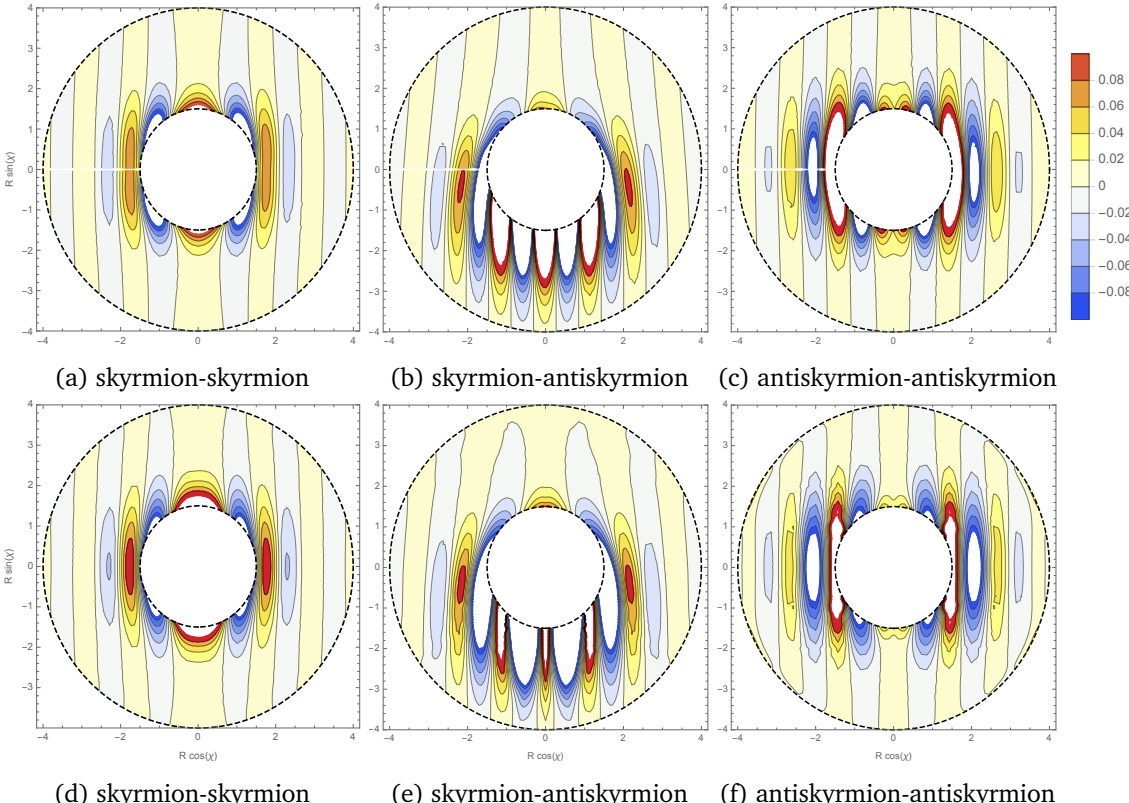

Figure 6: Analytical (a,b,c) and numerical [22] (d,e,f) results for the interaction potential between solitons in a tilted field, as a function of their separation. Dashed lines represent the limits of the numerical antiskyrmion-antiskyrmion interaction data, which is the most restrictive.

shutter effect [44], where rotational motion of a some object at angular speed $\omega$ is combined with the lateral motion of a camera shutter (at speed $v$) to produce an image where the object is distorted. This distortion can be described in terms of a map on polar co-ordinates [45]:

$$(r, \phi) \mapsto \left( r, \phi - \frac{\omega}{v} r \cos \phi \right). \tag{61}$$

We can see that each term in (55) where $M \neq N$ looks like the distorted image of the potential

$$2\pi(-1)^{N+1} q_M^A q_N^B \cos((N-M)\phi_0 + (M-N)\chi) K_{|M-N|}(mR), \tag{62}$$

if it were rotating at $\frac{\omega}{v} = \frac{a}{N-M}$. If one were to plot this function, it would look like a propellor with $|M-N|$ blades, hence the likeness between this interaction potential and the photo of rolling shutter effect in Fig. 7. The combination of all terms with $M \neq N$, then, is analogous to a photo of a combination of propellors with different numbers of blades spinning at different speeds, but the dominant multipoles will have the largest contribution. In our case, the effective propellor is dominated by the $M = 0$, $N = -1$ term, together with the $M = 1$, $N = 0$ term which is proportional to it and 'rotates' at the same speed. Meanwhile, the terms with $M = N$ all add together to give a term proportional to $\cos(-aR \cos \chi)$, i.e. oscillation only along $\vec{a}$. The combination of these two parts gives us the interaction potential above.

Between two identical solitons, equal contributions from oppositely rotating propellors will add up and obscure the rolling shutter picture, as they must to retain the $\chi \to \chi + \pi$ symmetry, but for interactions between unlike solitons we can see this will be a general feature, assuming only a small number of multipole sources contribute significantly as is seen here. This rolling

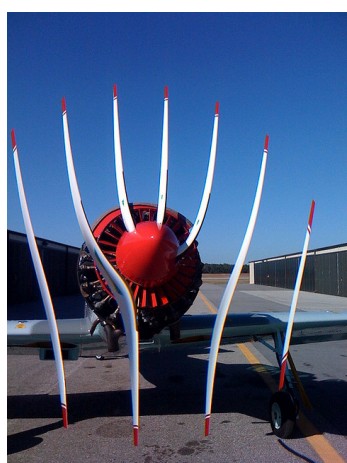

Figure 7: "Airplane Prop + CMOS Rolling Shutter = WTF", Soren Ragsdale, licenced under CC BY 2.0

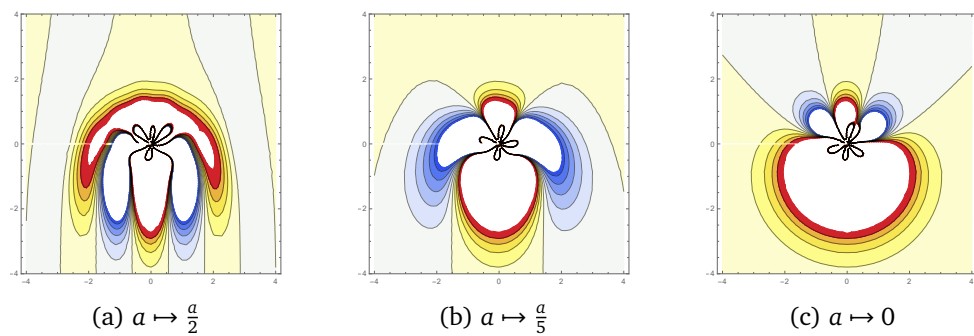

(a) $a \mapsto \frac{a}{2}$                 (b) $a \mapsto \frac{a}{5}$                 (c) $a \mapsto 0$

Figure 8: The skyrmion-antiskyrmion interaction potential for different values of $a = k \sin \theta_h$, where $k$ is the DMI strength and $\theta_h$ is the applied field tilt. At the same time we leave other parameters $m$, $q_M^{Sk}$, $q_M^{ASk}$, $\gamma_M^{Sk}$, $\gamma_M^{ASk}$ constant. This models how the interaction potential will change as the field tilt is reduced, but it is only schematic since in reality the parameters we are fixing are also functions of $\theta_h$. In particular, as $\theta_h \to 0$ then $q_{M \neq 1}^{Sk} \to 0$ and $q_{M\,\mathrm{even}}^{ASk} \to 0$, as both the skyrmion and antiskyrmion regain symmetry. The interaction potential would accordingly gain a second reflection symmetry about the x axis.

shutter analogy also gives us an idea of how the potential will change as we take $\theta_h \to 0$: the lobes of positive and negative interaction energy will move upwards as they become 'less distorted'. This change is illustrated qualitatively in Fig 8, where we take $a = k \sin \theta_h$ to zero while leaving all other parameters constant.

## 3.4 Soliton interactions in axisymmetric potential

In the case of axisymmetric potential and axisymmetric DMI, there are two possibilities: either a soliton retains the $U(1)$ symmetry of the energy (7) and thus has no internal degrees of freedom, or it breaks it and has one internal degree of freedom coming from its orientation. We call these degrees of freedom $\phi_0^A$, $\phi_0^B$ for the corresponding solitons. By considering the action of this $U(1)$ symmetry on a soliton, we can see that $q_M^{A,B}$ will be independent of $\phi_0^{A,B}$ while $\gamma_M^{A,B}$ will have a prescribed dependence.

We can generally define these orientations up to $\pm\pi$ by considering the dispersion tensor $\Gamma_{ij} = \int \partial_i \boldsymbol{n} \partial_j \boldsymbol{n} d^2 x$ [46]: when this matrix is not proportional to the identity matrix, it has a

one-dimensional eigenspace corresponding to its largest eigenvalue, picking a direction in the plane. The angle between this line and the $x$-axis defines $\phi_0^{A,B}$. When the soliton has at least one reflection-like symmetry (40), $\phi_0^{A,B}$ is also the angle of one of the spatial reflections, and the $\gamma_M^A, \gamma_N^B$ can be expressed in terms of $\phi_0^{A,B}$ according to (41). Note that the $\gamma_M^A, \gamma_N^B$ will not see the $\pm\pi$ ambiguity.

Considering also that $a = 0$ (this also means that $\tilde{\psi}_{\text{lin}} = \psi_{\text{lin}}$), the general formula (55) becomes

$$V_{\text{lin}}^{AB}(R, \chi; \phi_0^A, \phi_0^B) = 2\pi \sum_{M,N} (-1)^{N+1} q_M^A q_N^B \cos(\gamma_M^A(\phi_0^A) - \gamma_N^B(\phi_0^B) + (M-N)\chi) K_{|M-N|}(mR), \quad (63)$$

where we now make explicit that $\gamma_M^{A,B}$ depend on $\phi_0^{A,B}$ respectively, while $q_M^{A,B}$ do not.

According to (56), $V_{\text{lin}}^{AB}$'s dependence on $R$ and $\chi$ splits at sufficiently large $R$:

$$V_{\text{lin}}^{AB}(R, \chi; \phi_0^A, \phi_0^B) = \sqrt{2\pi^3} f(\chi; \phi_0^A, \phi_0^B) \frac{e^{-mR}}{\sqrt{mR}} + O\left(\frac{e^{-mR}}{(mR)^{\frac{3}{2}}}\right), \quad (64)$$

where

$$f(\chi; \phi_0^A, \phi_0^B) = \left(\sum_{M,N} (-1)^{N+1} q_M^A q_N^B \cos(\gamma_M^A(\phi_0^A) - \gamma_N^B(\phi_0^B) + (M-N)\chi)\right). \quad (65)$$

Therefore for sufficiently large $R$, $V_{\text{lin}}^{AB}$ decreases in absolute value as a function of $R$, so for given $\chi$ the attraction is either repulsive, if positive, or attractive, if negative.

In axisymmetric potential and DMI, the $O(2)$ symmetry of the skyrmion around the soliton centre sets all sources except $q_1$ to zero, and sets $\gamma_1 = \frac{\pi}{2} - \beta$, as discussed in Sec. 2.4. Thus setting $q_1^A = q_1^B =: q_1^{Sk}$ in this case, the skyrmion-skyrmion potential is

$$V_{\text{lin}}^{SkSk}(R, \chi) = 2\pi (q_1^{Sk})^2 K_0(mR). \quad (66)$$

With the approximation of $K_0(mr) \simeq \sqrt{\frac{\pi}{2}} \frac{e^{-mr}}{\sqrt{mr}}$, this is the potential derived in [21]. It is also in principle applicable to the interaction of skyrmionium with a skyrmion, or two skyrmioniums, as this soliton has the same symmetry, e.g.

$$V_{\text{lin}}^{SkSkm}(R, \chi) = 2\pi q_1^{Sk} q_1^{Skm} K_0(mR), \quad (67)$$

where $q_1^{Skm}$ is the corresponding dipole source of the far field of the skyrmionium with its centre defined as the point at which $\mathbf{n} = \mathbf{e}_3$. As discussed above, this requires us to define an unambiguous way to define the combined field of a skyrmion and skyrmionium at separation $\vec{R}$. In all these cases, the interaction is independent of $\chi$, and repulsive for any $R$.

However, other solitons are supported in axisymmetric DMI and potential [26]. For example, an antiskyrmion can be supported for a small range of coupling parameters. We label its effective sources $q_M^{ASk}, \gamma_M^{ASk}$. As discussed in Sec. 2.4, because it breaks the $O(2)$ symmetry of the energy to a $\mathbb{Z}_2 \times \mathbb{Z}_2$ subgroup, it has an orientation which is free to vary, which we here call $\phi_0^{ASk}$. The multipole orientations $\gamma_M^{ASk}$ are fixed in terms of this orientation:

$$\gamma_M^{ASk} = -\frac{\pi}{2} + \beta - (M-1)\phi_0^{ASk} + n_M \pi, \quad n_M \in \mathbb{Z}, \quad (68)$$

while $q_M^{ASk} = 0$ for $M$ even.

If we then consider the interaction of a skyrmion with an antiskyrmion, we find

$$V_{\text{lin}}^{SkASk}(R, \chi; \phi_0^{ASk}) = 2\pi q^{Sk} \sum_{N \text{ odd}} q_N^{ASk} \cos((N-1)(\phi_0^{ASk} - \chi) - n_M \pi) K_{|N-1|}(mR). \quad (69)$$

Here we see the dependence of $V^{AB}$ on an internal parameter, as discussed in Sec. 3.1. This potential is stationary with respect to variations of $\phi_0^{ASk}$ for $\phi_0^{ASk} = \chi, \chi + \frac{\pi}{2}$, but we cannot guarantee that these are the only critical values of $\phi_0^{ASk}$, nor say whether they are maxima or minima. However we can show that the interaction potential can be made negative for any $R, \chi$ by a suitable choice of $\phi_0^{ASk}$. To show this we rely on the link between winding of an individual tail and the sign of the overall interaction potential (48). If we assume $\psi_{\mathrm{lin}}^{ASk}(r, \phi; \phi_0^{ASk})$ is not equal to zero for all values of $r$ larger than the soliton core size, it must wind once clockwise around the origin in the complex plane as we vary $\phi$ from 0 to $2\pi$ for topological reasons, meaning the product of the fields above winds twice clockwise around the origin as we vary $\chi$ from 0 to $2\pi$. Equivalently, the product above winds twice anticlockwise around the origin as $\phi_0^{ASk}$ varies from 0 to $2\pi$. So we know that the interaction can be made negative by varying $\phi_0^{ASk}$. Therefore according to Equation (64), the antiskyrmion can be oriented so as to attract the skyrmion.

This observation can be extended more generally to the variety of soliton solutions in axisymmetric potential, provided we can extend the pinning procedure so as to unambiguously find an interaction between any two solitons. The general result confirms the observation made in [26], namely that any solution with chiral kinks on the outer domain wall, which necessarily has a zero-mode $\phi_0$ associated to the $U(1)$ symmetry of the energy (7), can be rotated so that it attracts another soliton, provided the tail does not reach the vacuum at any finite distance from the core. The argument generalises what goes before: as we vary $\phi$ from 0 to $2\pi$, $\psi_{\mathrm{lin}}(r, \phi; \phi_0)$ will wind $-N_{\mathrm{kink}} + 1$ times anticlockwise around the origin in the complex plane, where $N_{\mathrm{kink}}$ is the chiral kink number, as defined in the above paper, associated to the outermost domain wall of the soliton. Now we use the fact that since $\phi_0$ describes the $U(1)$ symmetry of the energy, we can link it to the variation of $\phi$: $\psi_{\mathrm{lin}}(r, \phi; \phi_0 + \alpha) = e^{-i\alpha} \psi_{\mathrm{lin}}(r, \phi + \alpha; \phi_0)$. This means that as we increase $\alpha$ from 0 to $2\pi$, $\psi_{\mathrm{lin}}(r, \phi; \phi_0 + \alpha)$ will wind $-N_{\mathrm{kink}}$ times. This means that as we vary $\phi_0$ in any potential involving this soliton, there will be $|N_{\mathrm{kink}}|$ regions of negative interaction potential and thus at sufficiently large $R$, the two solitons can be oriented to attract each other. If two solitons attract at arbitrary distance, it implies a bound state of the two, although we cannot guarantee stability of the combined configuration against collapse. Nevertheless, this general proof of attraction between solitons with chiral kinks can be seen as a partial explanation of why antiskyrmions are not found alone but with a large number of other magnetic solitons [26].

# 4 Conclusion

In this paper we presented a framework for calculating soliton interactions in general and applied it to get explicit formulae in two new cases: chiral magnetic skyrmions and antiskyrmions interacting in a tilted applied magnetic field, and a variety of magnetic solitons interacting in a chiral magnet with normally applied magnetic field and anisotropy. The treatment is general enough to include Bloch and Néel-type DMI. In the case of tilted field, we found close agreement between the analytical formula and previous numerical observations [22]. In the case of normal magnetic field combined with anisotropy, we found that solitons with chiral kinks [26] can always be oriented so as to attract another soliton.

This calculation generalised previous ones in the topological soliton literature by incorporating the effect of a $U(1)$ background gauge connection on the effective scalar field theory that describes the tails of solitons. This arises when considering tilted field applied to a chiral magnet, as the non-zero overlap between the minimum of the potential $\boldsymbol{n}_0$ and the DMI vectors $\boldsymbol{D}_i$ leads to a new term in the linearised Euler-Lagrange equations. This $U(1)$ gauge connection naturally descends from understanding the DMI as an $SO(3)$ gauge connection.

This then gives an explanation of the oscillating interaction potential generically observed in a tilted field.

We also generalised previous calculations in the magnetic skyrmion literature specifically, by understanding the far field of a general soliton as being effectively sourced by an infinite set of multipoles, rather than just a single one. This is required for any soliton that does not have the especially high symmetry of, for example, skyrmion and skyrmionium solutions. To predict interaction potentials, we therefore had to extract the strengths of these multipoles from looking at the isolated soliton. This sort of calculation has been done before in the context of nuclear skyrmions [24], although our methods here are different. This allowed us to consider interactions involving non-axisymmetric magnetic solitons like the antiskyrmion, which had not previously been calculated.

The discussion in this paper can be generalised in various directions. Firstly, the interaction formula (55) could be applied directly to any energy functional that satisfies the restrictions that the potential is isotropic close to the vacuum (10), and with a sufficiently large effective mass, (19). In particular, a case we have not considered in this paper is the same range of potentials (11), (12) but with general DMI that is not axisymmetric. Secondly, we could extend to cases where the potential does not satisfy the isotropy condition, as in (13), (14). The equations (48) and (51) would still be true, but even the linearised Euler-Lagrange equations cannot be solved exactly, only approximated [36]. Finally, the methods used can be applied to any case where exponentially localised configurations interact, in a chiral magnet or any similar medium. This includes skyrmion strings, Bloch points and more. For this the formula (B.11) could be used with the necessary modifications.

Separately, we used the study of linearised Euler-Lagrange equations of the chiral magnet, which were necessary to calculate the interaction potential, to investigate the magnetic soliton elliptical instability. The divergence of decay lengthscale of solutions to these equations coincides with the uniform state becoming linearly unstable, and we predict elliptical instability as we approach this region. The predictions closely match numerically observed elliptical instability when the applied field is tilted close to the plane. Remarkably, this instability calculation combines with a completely different method of estimating soliton instability, by finding the point at which domain walls become energetically favoured, to overall produce a good fit to to numerical observations at both small and large tilts of the applied field. It is interesting that a seemingly single phenomenon can be explained only by a combination of two such different methods. One can ask whether this reflects a real distinction between two forms of elliptical instability, or alternatively how these two methods are related, and if they can be combined.

## Acknowledgements

We thank Vlad Kuchkin for providing numerical data from [22].

**Funding information** B. B.-S. acknowledges an EPSRC-funded PhD studentship.

## A   Numerical method

Skyrmion and antiskyrmion solutions were generated by arrested Newton flow [47] on a 400x400 lattice with periodic boundary conditions, with grid size $dx = 0.02$. This means that the DMI lengthscale $\frac{1}{k}$ was approximately equal to $8dx$, and the size of the domain $400dx$ was approximately ten times $\frac{1}{m}$, the decay length. Exact solutions from the critically coupled model [31] with the appropriate charge were used as initial configurations.

To Fourier transform the angular dependence of $\tilde{\psi}$, we first must approximate the location of the soliton centre. This was done by looking for the lattice point where $\boldsymbol{n}$ was closest to $-\boldsymbol{e}_3$. This point then defines the centre of a polar co-ordinate system $(r, \phi)$. We then took a series of circles at successive values of $r$ and a range of $\phi$ and found $\tilde{\psi}$ by interpolation, then calculated the integral (42) for each $r$. The maximum radius at which this was done was half the distance to the edge of the domain. It was found that beyond this the effects of the finite size of the domain caused $|c_M(r)|/K_M(mr)$ to again deviate from a constant value.

This method of finding the soliton centre is fairly crude, and by using more sophisticated methods we can reduce the discrepancy between our measurement for $\gamma_M$ and the value predicted by (41). So to account for this discrepancy in general, we consider the error in the location of the soliton centre, $\delta x_0$, in the calculations below. This leads to errors in the polar co-ordinates ($\delta r = \delta x_0, \delta \phi = \frac{\delta x_0}{r}$). Note that these are not independent at each point. This leads to an error in the argument of $\tilde{\psi}$, $\delta \arg(\tilde{\psi}) = a \delta x_0$, and thus an error in $\arg(c_M(r))$. Then in the integral (42), the error in $\phi$ leads to an error in $\arg(c_M(r))$ equal to $M \delta x_0/r$, while the error in $r$ leads to an error in $|c_M(r)|$ equal to $c'_M(r)\delta x_0$. These are the errorbars plotted in Figs. 1, 2 and 5.

To test the accuracy of this method, we simulated the skyrmion in normal magnetic field, $\boldsymbol{D}_i = -2\pi \boldsymbol{e}_i, h_z = 0.8(2\pi)^2, h_a = 0, \theta_h = 0$. Because of the axisymmetry of the skyrmion, we expect only $C_1 \neq 0$. The largest $|C_{M \neq 1}|$ we found numerically was 0.07, compared to a value of $\sim 7.3$ for $|C_1|$. We can also compare the value of $|C_1|$ found to that for the hedgehog solution of the Euler-Lagrange equations as found by shooting. In Fig. 9, both solutions are shown to be well-fitted by the appropriate Bessel function $K_1(mr)$ beyond a certain radius, but there is a discrepancy between them that leads to a 1.9% error in the calculated value of $|C_1|$, which we see is well contained within the errorbars described above.

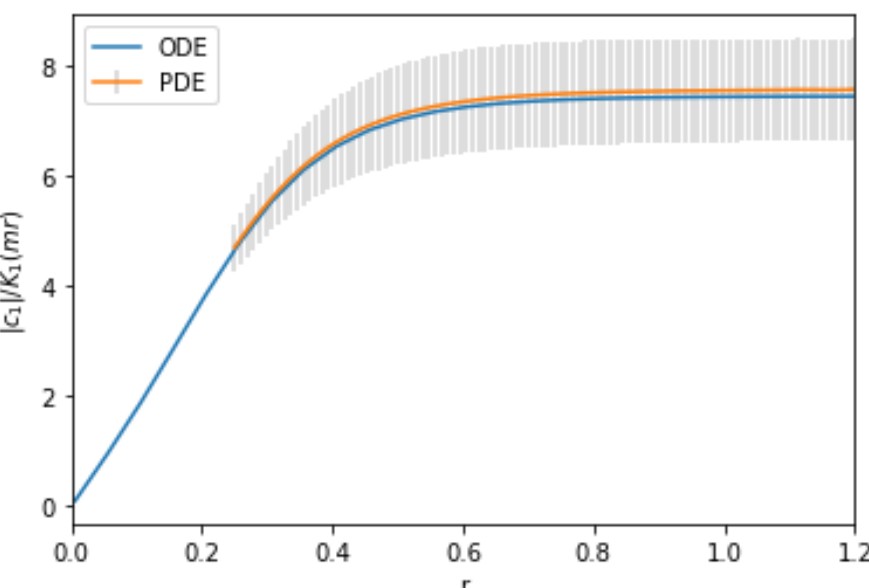

Figure 9: Magnitude of angular Fourier term $c_1(r)$ of the axisymmetric skyrmion tail relative to the Bessel function $K_1(mr)$, for the solution found by arrested Newton flow (PDE) and for the solution of the hedgehog Euler-Lagrange equations found by shooting (ODE), for the chiral magnet energy functional (1) with axisymmetric DMI (6) of strength $k = 2\pi$ and axisymmetric potential (11) $h_z = 0.8(2\pi)^2, h_a = 0$.

# B  Interaction potential for general energy functional

Here we discuss a general framework for calculating the interaction energy of solitons in a magnetisation field (i.e. target space $S^2$) with an arbitrary local energy functional, provided that solutions of the Euler-Lagrange equations fall off exponentially. The argument is in fact more general still. It applies for target spaces that are any Riemannian manifold, meaning that we can for instance consider the target space to be $SO(3)$, which can be relevant for certain classes of antiferromagnet. It also holds for any number of spatial dimensions, which would for instance be useful if we want to calculate the interaction strength of skyrmion strings. However, for the following we keep the number of spatial dimensions to two and the target space to $S^2$. The steps of this argument are not new [17,23], but the purpose of this discussion is to show its generality and that it does not require a superposition ansatz. Throughout we think in terms of the infinite-dimensional configuration space consisting of all possible configurations of the magnetisation field.

As described in Sec. 3.1, we define the interaction potential $V^{AB}$ in terms of the difference between the energy of a combined configuration $\boldsymbol{n}^{AB}$ and the energy of the two isolated soliton configurations $\boldsymbol{n}^A$, $\boldsymbol{n}^B$. In the simplest case, we construct $\boldsymbol{n}^{AB}$ as a function of $\vec{R}$ by finding the absolute minimum of the energy, within configurations of an appropriate topological degree, subject to the constraint that $\boldsymbol{n}^{AB} = -\boldsymbol{e}_3$ at $\vec{0}$ and $\vec{R}$. This creates a 2-dimensional moduli space within the infinite-dimensional configuration space, parametrised by $\vec{R}$. This is motivated by numerical pinning procedures to approximate interaction energies, with which we make direct comparison [22]. This moduli space also has a significance when the Landau-Lifshitz equation is dominated by damping, so that in mathematical terms fields undergo gradient flow on the configuration space. If we assume that under this dynamics the soliton centres will move together or apart while remaining unique, then this pinning procedure consists of foliating the configuration space transversally to the path of the true dynamics and finding the minimum on each sheet. In particular, this means that gradient flows that start within our moduli space will remain on it, so that the path of gradient flow on the moduli space is the path of the dynamics on the configuration space and so the moduli space approach is exact, not an approximation.

There are other choices that we could make to construct $\boldsymbol{n}^{AB}$. Firstly, our choice of soliton centre is somewhat arbitrary and we only choose the point at which $\boldsymbol{n} = -\boldsymbol{e}_3$ for comparison to numerics, as discussed in the main paper. A more natural choice might be the point $\vec{x}_0$ at which $\boldsymbol{n}^{AB}(\vec{x}_0) = -\boldsymbol{n}_0$, as this continues to be unique as we rotate the groundstate from $\boldsymbol{e}_3$ to $-\boldsymbol{e}_3$. The choice of soliton centre can also be on a case-by-case basis for different solitons. As discussed in the main paper, we assume that in general we can modify the constraints to specify between all the soliton configurations under consideration.

A separate approach, if we are interested in the property that the moduli space contains gradient flows of the full energy, is to explicitly construct it as such a space. To do this we extend our space of configurations to include the solitons at infinite separation, and look at the (un)stable manifold of this configuration if the attraction is repulsive (attractive) [48]. Another approach would be to construct $\boldsymbol{n}^{AB}$ by ansatz, namely some pointwise superposition of the two fields [17, 23], that satisfies the properties we require below. This would give an upper bound on the interaction energy as defined by the pinning method, if it were calculated exactly, as in [49].

For the purposes of this appendix, the composite field $\boldsymbol{n}^{AB}$ only has to satisfy a few basic features in terms of the fields $\boldsymbol{n}^A$, $\boldsymbol{n}^B$ and $\vec{R}$. Firstly, it should attain $-\boldsymbol{e}_3$ at $\vec{0}$ and $\vec{R}$. Secondly, as $R \to \infty$, $\boldsymbol{n}^{AB} \to \boldsymbol{n}^A$ around soliton $A$, and $\boldsymbol{n}^{AB} \to \boldsymbol{n}^B$ around soliton $B$. Thirdly, the field and its derivatives fall off exponentially away from the soliton core. Fourthly, the fields approach linear superposition far from both soliton cores, as discussed in the introduction. We can make these assumptions more precise below once we have introduced some notation.

We define the region closer to $\vec{0}$ than $\vec{R}$ as $\sigma^A$, and its complement as $\sigma^B$. Since the energy functional is local, we can write the energy functional $E$ as the sum of the integrals over the two regions, $E = E_{\sigma^A} + E_{\sigma^B}$. We can now split the calculation for $V^{AB}$:

$$V^{AB} = E_{\sigma^A}\left(\boldsymbol{n}^{AB}\right) - E_{\sigma^A}\left(\boldsymbol{n}^A\right) - E_{\sigma^A}\left(\boldsymbol{n}^B\right) + E_{\sigma^B}\left(\boldsymbol{n}^{AB}\right) - E_{\sigma^B}\left(\boldsymbol{n}^A\right) - E_{\sigma^B}\left(\boldsymbol{n}^B\right). \tag{B.1}$$

To continue we call a map $\boldsymbol{\epsilon} : \mathbb{R}^2 \to \mathbb{R}^3$ a tangent vector to a given magnetisation field $\boldsymbol{n}$ in configuration space if $\boldsymbol{\epsilon}(\vec{x}) \cdot \boldsymbol{n}(\vec{x}) = 0$ for all $\vec{x}$. Using the exponential map on the sphere defined in the main text (8), we define the exponential map in configuration space [50] $\exp_{\boldsymbol{n}}$ which takes a tangent vector $\boldsymbol{\epsilon}$ to $\boldsymbol{n}$, and turns it into a magnetisation field that is pointwise the exponential map $\exp_{\boldsymbol{n}(x)}(\boldsymbol{\epsilon}(x))$:

$$(\exp_{\boldsymbol{n}}(\boldsymbol{\epsilon}))(\vec{x}) = \exp_{\boldsymbol{n}(\vec{x})}(\boldsymbol{\epsilon}(\vec{x})). \tag{B.2}$$

In addition to $\boldsymbol{\psi}_{\boldsymbol{n}^B}$ as defined in the main text (9), we can also define $\boldsymbol{\epsilon}^B = \exp_{\boldsymbol{n}^A}^{-1}(\boldsymbol{n}^{AB})$. The first describes the 'tail' of the isolated soliton $B$, while the second describes the perturbation of the tail of soliton $B$ on soliton $A$. As shown in 2.3, $\boldsymbol{\psi}_{\boldsymbol{n}^B}$ can be approximated by solving the linearised Euler-Lagrange equations for the particular energy functional we are working with. Meanwhile, we have no explicit way of finding $\boldsymbol{\epsilon}^B$ but we can link it to $\boldsymbol{\psi}_{\boldsymbol{n}^B}$ using the assumptions on $\boldsymbol{n}^{AB}$.

Note that according to (9), $\exp_{\boldsymbol{n}_0}^{-1}(-\boldsymbol{n}_0)$ is not defined, so we can only define $\boldsymbol{\psi}_{\boldsymbol{n}^{A,B}}(\vec{x})$, $\boldsymbol{\epsilon}^{A,B}(\vec{x})$ when $\boldsymbol{n}(\vec{x})$ is never $-\boldsymbol{n}_0$. However, $\boldsymbol{n}^{AB}(\vec{x})$ does indeed reach $-\boldsymbol{n}_0$ near or at the soliton centre. This is not a problem in practice: note that we always define $\boldsymbol{\psi}_{\boldsymbol{n}^{A,B}}(\vec{x})$ and $\boldsymbol{\epsilon}^{A,B}(\vec{x})$ below in integrals over regions away from the corresponding soliton centre. In actuality we are then using the fact that the integral is independent of the value of $\boldsymbol{n}^{AB}(\vec{x})$ outside its domain to replace $\boldsymbol{n}^{AB}(\vec{x})$ with a field that is identical within the domain of integration but outside the integral never reaches $-\boldsymbol{n}_0$, and then defining $\boldsymbol{\epsilon}^B$, $\boldsymbol{\psi}_{\boldsymbol{n}^B}$ in terms of the inverse exponential function on that field. We just skip this cumbersome notation.

We can now state our second assumption on $\boldsymbol{n}^{AB}$ more explicitly: as we go far from the centre of soliton $B$, $|\boldsymbol{\epsilon}^B(\vec{x})| \sim |\boldsymbol{\psi}_{\boldsymbol{n}^B}(\vec{x})|$, and thus $|\boldsymbol{\epsilon}^B(\vec{x})| \to 0$ as $R \to \infty$ for all $\vec{x}$ in region $\sigma^A$, and vice versa. We can also formalise our assumption of linear superposition: in the region far from both solitons, $\boldsymbol{\epsilon}^{A,B} \to \boldsymbol{\psi}_{\boldsymbol{n}^{A,B}}$. Finally, we have the assumption that the tails and their derivatives are bounded by exponential decay: $\boldsymbol{\psi}_{\boldsymbol{n}^{A,B}}(\vec{x}) = O(e^{-mr})$, $\partial_i \boldsymbol{\psi}_{\boldsymbol{n}^{A,B}}(\vec{x}) = O(e^{-mr})$, and so on. This last assumption is satisfied by solitons in the chiral magnet, see Equations (29), (32), but in this appendix the discussion is more general and $m$ can be taken to generally describe the inverse decay lengthscale of whatever soliton is under consideration.

We define the variation of the energy functional:

$$\delta_{\boldsymbol{\epsilon}} E(\boldsymbol{n}) = \frac{d}{dt}\bigg|_{t=0} (E(\exp_{\boldsymbol{n}}(t\boldsymbol{\epsilon})), \tag{B.3}$$

and the second variation:

$$\delta_{\boldsymbol{\epsilon}', \boldsymbol{\epsilon}}^2 E(\boldsymbol{n}) = \frac{d}{dt}\bigg|_{t=0} \delta_{\boldsymbol{\epsilon}} E(\exp_{\boldsymbol{n}}(t\boldsymbol{\epsilon}')), \tag{B.4}$$

then we can Taylor expand all three terms:

$$E_{\sigma^A}(\boldsymbol{n}^{AB}) - E_{\sigma^A}(\boldsymbol{n}^A) = \delta_{\boldsymbol{\epsilon}^B} E_{\sigma^A}(\boldsymbol{n}^A) + \frac{1}{2}\delta_{\boldsymbol{\epsilon}^B, \boldsymbol{\epsilon}^B}^2 E_{\sigma^A}(\boldsymbol{n}^A) + O\left(|\boldsymbol{\epsilon}^B|^3\right), \tag{B.5}$$

$$E_{\sigma^A}(\boldsymbol{n}^B) = E_{\sigma^A}(\boldsymbol{n}_0) + \delta_{\boldsymbol{\psi}_{\boldsymbol{n}^B}} E_{\sigma^A}(\boldsymbol{n}_0) + \frac{1}{2}\delta_{\boldsymbol{\psi}_{\boldsymbol{n}^B}, \boldsymbol{\psi}_{\boldsymbol{n}^B}}^2 E_{\sigma^A}(\boldsymbol{n}_0)) + O\left(|\boldsymbol{\psi}_{\boldsymbol{n}^B}|^3\right)$$

$$= \delta_{\boldsymbol{\psi}_{\boldsymbol{n}^B}} E_{\sigma^A}(\boldsymbol{n}_0) + \frac{1}{2}\delta_{\boldsymbol{\psi}_{\boldsymbol{n}^B}, \boldsymbol{\psi}_{\boldsymbol{n}^B}}^2 E_{\sigma^A}(\boldsymbol{n}_0) + O\left(|\boldsymbol{\psi}_{\boldsymbol{n}^B}|^3\right), \tag{B.6}$$

where $|\epsilon|$ is the maximum value of $|\epsilon(x)|$ over the whole region $\sigma^A$, in practice its value at the midpoint $\vec{x} = \frac{1}{2}\vec{R}$. We use our assumption that $|\epsilon| \sim |\psi|$ to replace $O(|\epsilon^B|^3)$ with $O(|\psi_{n^B}|^3)$, and so on.

This is where the exponential decay assumption becomes useful: since we are expanding in terms of the maximum value of $|\psi_{n^B}(x)|$ over the domain of integration, this expansion only makes sense if in general the integral of the function is bounded by its maximum value, and whatever derivative operators may act within the integrals that we throw away do not change the order of that maximum value. Both of these things are true for exponentially decaying functions, and thus true for functions bounded below exponential decay with all derivatives bounded below exponential decay. We will use this several times more below to bound integrals in terms of the maximum value of a field over the domain of integration. At this point we can bound the term we are neglecting in terms of $\max_{\sigma^A} e^{-3mr} = e^{-\frac{3}{2}mR}$.

For what follows it is useful to separate the first variation into two contributions. By integration by parts, a variation can always be separated into a bulk integral where the variation field appears without derivatives, and an integral along $\partial\sigma$ which we call $\partial\delta_\epsilon E_\sigma(\boldsymbol{n})$:

$$\delta_\epsilon E_\sigma(\boldsymbol{n}) = \int_\sigma \epsilon \cdot \mathbf{f}(\boldsymbol{n}, \partial\boldsymbol{n}, \ldots)d^2x + \partial\delta_\epsilon E_\sigma(\boldsymbol{n}). \tag{B.7}$$

Because $\boldsymbol{n}^A$ is a minimizer of the full energy $E(\boldsymbol{n})$, $\boldsymbol{n}^A$ solves the Euler-Lagrange equations, $\boldsymbol{n} \times \mathbf{f}(\boldsymbol{n}, \partial\boldsymbol{n}, \ldots) = \mathbf{0}$, and therefore the term $\delta_{\epsilon^B} E_{\sigma^A}(\boldsymbol{n}^A)$ can only be a boundary term:

$$\delta_{\epsilon^B} E_{\sigma^A}(\boldsymbol{n}^A) = \partial\delta_{\epsilon^B} E_{\sigma^A}\left(\boldsymbol{n}^A\right). \tag{B.8}$$

Similarly, $\delta_{\psi_{n^B}} E_{\sigma^A}(\boldsymbol{n}_0) = \partial\delta_{\psi_{n^B}} E_{\sigma^A}(\boldsymbol{n}_0)$. Now because $\partial\delta_{\epsilon^B} E_{\sigma^A}(\boldsymbol{n}^A)$ depends only on the field $\boldsymbol{n}^A$ evaluated along $\partial\sigma^A$, which is far from $\vec{0}$, we can Taylor expand the whole expression around $\boldsymbol{n}^A = \boldsymbol{n}_0$, using the fact that $\epsilon^B \to \psi_{n^B}$:

$$\partial\delta_{\epsilon^B} E_{\sigma^A}(\boldsymbol{n}^A) = \partial\delta_{\psi_{n^B}} E_{\sigma^A}(\boldsymbol{n}_0) + \delta_{\psi_{n^A}}\partial\delta_{\psi_{n^B}} E_{\sigma^A}(\boldsymbol{n}_0) + O\left(e^{-\frac{3}{2}mR}\right). \tag{B.9}$$

This means that

$$V^{AB} = \delta_{\psi_{n^A}}\partial\delta_{\psi_{n^B}} E_{\sigma^A}(\boldsymbol{n}_0) + \frac{1}{2}\delta^2_{\epsilon^B,\epsilon^B} E_{\sigma^A}(\boldsymbol{n}^A) - \frac{1}{2}\delta^2_{\psi_{n^B},\psi_{n^B}} E_{\sigma^A}(\boldsymbol{n}_0) + (A \leftrightarrow B) + O\left(e^{-\frac{3}{2}mR}\right). \tag{B.10}$$

Outside the soliton core, we can expand $\frac{1}{2}\delta^2_{\epsilon^B,\epsilon^B} E_{\sigma^A}(\boldsymbol{n}^A)$ around $\boldsymbol{n}_0$, and the latter two terms will cancel to order $|\epsilon^B|^2|\psi_{n^A}|'$, where $|\psi_{n^A}|'$ is the maximum value of $|\psi_{n^A}(\vec{x})|'$ over everywhere except the soliton core. This is still subleading. Inside the soliton core, both terms are order $|\epsilon^B|'^2$, where $|\epsilon^B|'$ is the maximum value of $|\epsilon^B(\vec{x})|$ over just the soliton core. Because of our assumption that $\boldsymbol{n}$ approaches $\boldsymbol{n}^A$ in $\sigma^A$ as $R$ increases, the size of the soliton core approaches a constant value, so it can be arbitrarily small in comparison to $R$, and so in particular the distance between the centre of soliton $B$ and the closest edge of the core of soliton $A$ can be larger than $\frac{3R}{4}$ for large enough $R$. Then $|\epsilon^B|'^2 \sim |\epsilon^B|^3 = O\left(e^{-\frac{3}{2}mR}\right)$ and we get our final expression for the general interaction energy at large separation:

$$V^{AB} = \delta_{\psi_{n^A}}\partial\delta_{\psi_{n^B}} E_{\sigma^A}(\boldsymbol{n}_0) - \delta_{\psi_{n^B}}\partial\delta_{\psi_{n^A}} E_{\sigma^A}(\boldsymbol{n}_0) + O\left(e^{-\frac{3}{2}mR}\right), \tag{B.11}$$

where we use the fact that $\partial\sigma^B$ is equal to $\partial\sigma^A$ with the opposite orientation.

We find in the main text that $V^{AB} \sim \frac{e^{-mR}}{\sqrt{mR}}$, so the correction term is indeed subleading. By repeating the above calculations in a more geometric language, we can view the term $\partial\delta_\epsilon E_{\sigma^A}$ as a one-form on the configuration space of magnetisation fields, acting on the vector $\epsilon$, $\partial\delta_\epsilon E_{\sigma^A}(\boldsymbol{n}^A) = \omega_{n^A}(\epsilon)$. Then our final expression can be seen to be the external derivative of this one-form acting on the two tail vector fields, $V^{AB} = d_{n_0}\omega(\psi_{n^A}, \psi_{n^B})$.

We can apply this formula to the chiral magnet energy functional (1). First we find the boundary term from the first variation of the energy:

$$\partial\,\delta_{\psi}E_{\sigma^A}(\boldsymbol{n}) = \int_{\partial\sigma^A} \boldsymbol{\psi}\cdot(\partial_i\boldsymbol{n} + \boldsymbol{D}_i\times\boldsymbol{n})dS^i\,, \tag{B.12}$$

then we vary this term with respect to a second field:

$$\delta_{\psi'}\partial\,\delta_{\psi}E_{\sigma^A}(\boldsymbol{n}_0) = \int_{\partial\sigma^A} \boldsymbol{\psi}\cdot(\partial_i\boldsymbol{\psi}' + \boldsymbol{D}_i\times\boldsymbol{\psi}')dS^i\,. \tag{B.13}$$

Note that the addition of a boundary term to the energy like $-\boldsymbol{D}_i\cdot(\boldsymbol{n}_0\times\partial_i\boldsymbol{n})$, which can be motivated physically and in terms of analysis [30, 32], does not affect this quantity and thus does not enter into the interaction potential. Also, while for simplicity we chose $\partial\sigma^A$ to be the straight line equidistant from the soliton centres, the derivation above only fundamentally depends on the fact that $\partial\sigma^A$ divides the plane into two halves with a soliton core in each half. We see this independence of the exact boundary in the main section when we calculate the interaction potential for the chiral magnet specifically. Finally we substitute this into (B.11) to find (47).

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
