# Peer review of "Stability and asymptotic interactions of chiral magnetic skyrmions in a tilted magnetic field"

_SciPost Physics, doi:SciPost Phys. 15, 011 (2023)_

## Round 1 · Referee Report · Anonymous (Referee 1) · 2023-2-15

Strengths

  1. Well written.
  2. The main message of the work is delivered correctly.
  3. Broad overview of the problem with appropriate selection of references.
  4. Precise formulations.
  5. Very detailed description of the primary result derivation.

Weaknesses

  1. One essential reference is missing
  2. Minor comments explaining figure 8 should be added.

Report

The interaction between magnetic solitons is a timely and interesting topic, and the present work by Barton-Singer and Schroers makes an important contribution to this field. Their study proposes an elegant and reliable approach for analytically modeling the interaction potential between magnetic solitons. The manuscript is well-organized and well-written.

I have two minor comments on the content of the paper:

  1. In the introduction section, the authors wrote: "We discuss the interaction of novel textures that have recently been numerically observed in magnetic field applied normal to the plane [21,25]." I'm sure that the authors would agree that a reference to PRB 99, 064437 (2019) is missing here.

  2. In Figure 8, the authors provide a qualitative illustration of the reduction in external magnetic field tilt. In a perpendicular field, the skyrmion becomes axially symmetric, and the antiskyrmion has second-order rotational symmetry. The interaction potential between the skyrmion and antiskyrmion should also have second-order symmetry, similar to Fig. 6 (d) and (f). Instead, in Fig. 8c, we see that the potential remains distorted, as in the case of the first-order symmetry distortion of the skyrmion and antiskyrmion. The authors should add a comment on this effect, which might be confusing for readers.

Overall, the presented work fully meets the acceptance criteria for the SciPost Physics journal.

Requested changes

  1. Add a reference to PRB 99, 064437 (2019).
  2. Explain the broken symmetry of interaction potential in Fig.8 c.

---

## Round 1 · Referee Report · Anonymous (Referee 2) · 2023-3-1

Strengths

1 Addresses a problem of strong current interest, and makes substantial progress on it.
2 Very clearly written.
3 Links well to, and expands on, existing results in the literature.

Weaknesses

1 Sometimes difficult to keep track of which precise variant of the model is under discussion.

Report

This is a well-written and high quality paper addressing an important problem. The main ideas are clearly explained, but the main text is not overloaded with detailed mathematical derivations because the authors have made judicious use of appendices. The authors have emphasized field-theoretic interpretations and analogues of their results which is a helpful approach.

Minor criticisms: 1) Some of the figure captions are difficult to understand in isolation. For example, figure 1 refers to several unspecified parameters which are quite difficult to dig out from the text (k, in particular, requires one to remember material from several pages previously). In general, figure captions should be comprehensible without excessive digging. The caption for figure 3 also requires some proof reading.

2) The phrase "the integral of the gauge connection between the two sources" is a bit mysterious. I know what the authors mean: they're interpreting exp(-iaR) as the parallel transport map for the connection along a curve from source A to source B. But coming directly after (50), a formula involving integrals over two-dimensional regions, this meaning is a bit obscure.

3) Could the authors explain why they've consistently defined the soliton centre to be the point where n = (0,0,-1)? Wouldn't the point where n=-n_0 be more natural? This is where the perturbation field \psi is maximal. I'm not suggesting they rehearse their computations with this choice, but I would expect to see the point addressed.

Requested changes

1) Reword figure captions to make them more self-contained.

2) Clarify the meaning of "integral of the gauge connection".

3) Discuss the choice of soliton centre, in particular, why the (IMO) more natural choice n(x)=-n_0 was not used.

---

## Round 2 · Referee Report · Anonymous (Referee 1) · 2023-4-4

Report

After minor revisions and corrections made by the authors, the manuscript is now ready for publication.

---

## Round 2 · Referee Report · Anonymous (Referee 2) · 2023-5-3

Strengths

As 1st report

Weaknesses

None

Report

The authors have adequately addressed the issues I raised. I recommend publication of the revised version.

Requested changes

N/A

---

## Round 2 · Author Response

We thank the reviewers for their clearly careful consideration of the paper, and a number of useful comments. We have followed all of them.

---

## Round 2 · List of Changes

Reference to PRB 99, 064437 (2019) was added in the introduction.

A comment was added in the caption to Figure 8, explaining what is happening in more detail and to what extent this reflects the behaviour of the interaction potential.

All figure captions were revised to improve clarity, and in particular to allow them to stand alone from the text more.

The points 0 and R were added to Figure 4.

The paragraph involving the phrase 'the integral of the gauge connection between the two sources' was re-worded - the concept is referred to more clearly in the previous paragraph.

At the first introduction of the notion of soliton centre, the reasoning for using -e3 rather than -n0 is given. This was previously a comment in the appendix, which has been amended accordingly.

---

## Editorial Decision

published